# Mapping anthropogenic mineral generation in China and its implications for a circular economy

Xianlai Zeng [1,2], Saleem H. Ali [3,4,5], Jinping Tian [1] & Jinhui Li [1✉]

Anthropogenic mineral is absorbing wide concern in the context of circular economy, but its generation mechanism and quantity from product to waste remain unclear. Here we consider three product groups, 30 products, and use the revised Weibull lifespan model to map the generation of anthropogenic mineral and 23 types of the capsulated materials by targeting their evolution from 2010 to 2050. Total weight of anthropogenic mineral on average in China reached 39 Mt in 2010, but it will double in 2022 and quadruple in 2045. Stocks of precious metals and rare earths will increase faster than most base materials. The total economic potential in yearly-generated anthropogenic mineral is anticipated to grow markedly from 100 billion US$ in 2020 to 400 billion US$ in 2050. Furthermore, anthropogenic mineral of around 20 materials will be capable to meet projected consumption of three product groups by 2050.

[1] State Key Joint Laboratory of Environment Simulation and Pollution Control, School of Environment, Tsinghua University, 100084 Beijing, China.
[2] Center for Industrial Ecology, School of Forestry and Environmental Studies, Yale University, New Haven, CT 06511, USA. [3] College of Earth, Ocean and Environment, University of Delaware, Newark, DE 19709, USA. [4] Sustainable Minerals Institute, University of Queensland, Brisbane, Queensland 4072, Australia. [5] United Nations International Resource Panel, United Nations Environment Programme, Nairobi, Kenya. ✉email: jinhui@tsinghua.edu.cn

There is wide recognition that humans have indelibly affected planetary processes to merit current times being termed the Anthropocene[1,2]. The recent rapid expansion of high-tech industries[3,4], along with manufacturing innovations and consumer demand, have revolutionized societal investments in infrastructure for networking and for the rapid expansion of international commerce. However, the shortening useful life expectancy of the product, driven by rapid innovation, miniaturization and affordability, and an increasingly anthropogenic metabolism have led to a major increase in the accumulation of product waste, which could potentially be classified as anthropogenic mineral (AM, some relevant terminologies definition and boundaries are provided in Supplementary Information Note 1)[3,5]. One such waste stream is strictly regulated by the Chinese government. It is referred to as AM and was defined by the National Development and Reform Commission in 2010 to comprise the iron, non-ferrous metals, precious metals, plastic, or rubber material found within three types of product waste: waste electrical and electronic equipment (WEEE, all the abbreviations and acronyms are also provided at Supplementary Information Note 2), end-of-life vehicle (ELV), and waste wire and cable (WWC)[6,7]. These are identified as the core scope of AM not only in China, but also in many industrial nations[8–11].

Critical raw materials have also been sinking into AM reserves, while accelerating the depletion of natural minerals[12–15]. AM is expected to play an increasing role in resource supply. On the other hand, China is not only the world's major manufacturing power, but also one of the largest consumers and exporter of products[16]. Nowadays China has all types of industries in UN International Standard Industrial Classification System and led in the production of 220 of 500 global industrial products[17]. China in recent years was also the largest importer of secondary resource to alleviate domestic material scarcity. Such features could highlight the unique opportunity for China to uncover the potential of AM supply[18,19].

EEEs and vehicles are the most fashionable aspiration of assets in Chinese households, which are the hallmark of the Four Big Items that consumers are aspiring towards (see Supplementary Fig. 2). Their rapid evolution and popularity since the 1970s have led to a dramatic rise in waste accumulation and resource consumption. The consumption of some mineral resources has witnessed multiple increases[20], resulting in a shortage of important strategic resources and a growth of external dependence[21–23]. To meet future resource consumption, mining from AM has become a global concern and raised the popularization of the concept of a circular economy.

How to measure the quantity of AM generation and its role in future resource supply is still a crucial scientific challenge. Basic AM information—including generation, composition, and resource flow—is imperative to formulate effective policies making for the recycling industry. At least two gaps can be found in previous studies. First, there is a lack of full discussion for the quantity and quality of China's whole AM reserve and instead only individual types of waste streams have been considered. Zeng et al. (2016) measured the quantity and quality of e-waste, and uncovered China's recycling potential[24]. For ELV, van Schaik and Reuter defined the obsolescence rate and recycling rate in the EU[25], and Field et al. (2017) initialed a comprehensive assessment of strategic and minor metals use for passenger cars and light trucks[26]. Xue et al. (2013) established the discarding model to examine the recycling potential of ELV[27]. Furthermore, the other gap is no publication to accurately measure AM supply meeting potential for the future consumption. The dynamic transfer of existing resources from in-use stock to waste increases both the need for, and possibility of, sustainable resource harvesting from AM[28–30].

Urban metabolism is devoted to facilitate the analysis of the flows of the materials and energy within cities[31]. Theoretically, the generation and quantity of AM are subject to urban metabolism affected by a variety of regulations, resultant policy, and technological change[32,33]. However, due to challenges in tracking and recording, the accurate estimation of AM remains difficult to obtain[34]. To complete this study, we collect all the available data and initially create the mathematical models of AM recycling and meeting potential. Four procedures are employed: data collection for the consumption, importation, exportation, and material composition of AM; method development based on material flow analysis (MFA); generation estimation of AM quantity, and resultant economic potential; and validation of results using comparison with previous studies or reported data, sensitivity analysis, and uncertainty analysis (the detailed technical route can be seen in Fig. 1 and Supplementary Fig. 3).

## Results

**AM generation**. We firstly pre-mined the available data for further estimation of domestic generation of AM: First, lifetime distribution function of all the relevant products is determined by Supplementary Text 3 and Table 13; Second, data regression is enabled with Supplementary Figs. 7-9 for the annual net production and imported scraps until 2050. In this year China was attempted to become one moderately developed country; and last, the incremental error or range are configured for the estimation of future demand and importation (Supplementary Tables 11, 14). Accordingly, domestic generation of WEEE, ELV, and WWC are uncovered (Supplementary Fig. 10 and Fig. 2). The weight of WEEE on average was 4.67 million tons (1 Mt = 1000 kt = $10^6$ tons; 1 ton = $10^3$ kg) in 2010, but it will reach 27.22 Mt in 2030 and 51.60 Mt in 2050. The distribution of WEEE types in China is also evolving in its economic and material waste profile through the period 2010–2050. In 2015, the four types, by weight, were AC, DPC, monitor, and RF, accounting for 73% of the total WEEE. However, in 2020, the four types, by weight, will be DPC, AC, RF, and monitor, accounting for 76% of total WEEE. In addition, by 2030, the four types are expected to be AC, DPC, RF, and WM, accounting for 81% of total WEEE. The weight share of the remaining WEEE types has been shrinking since 2010.

For ELV, a continued increase also can be found both in weight and in quantity (Fig. 2). Actually, since 2009 China has become the largest generator and seller of vehicles. In the end of 2015, the total registered vehicles in service reached 279 million (M)[35], exceeding 264 M of the U.S.[36]. The total ELV reached 11.01 Mt in 2010, and three-fold rise will occur in 2017. Later, it will reach around 61.37 Mt in 2030 and 97.10 Mt in 2050. Thus, the average annual amount of ELV will be over 2 Mt in 2010–2050. Among the main large ELV, the share in 2010 was nearly 54%, 18%, 9%, 2%, and 0 for CT, PV, RV, tractor, and EV, respectively. With respect to WWC, those weights will remain under 5 Mt before 2025, and then dramatically increase to 33 Mt in 2040 (Fig. 2).

Regarding the importation, steel scrap, copper scrap, aluminum scrap, and plastics scrap are the majority of imported AM. In July 2017, China issued a rigorous policy of phasing out waste import, including some unrestricted import of waste (Supplementary Table 10)[37]. The linear fitting has been employed to model the imported e-waste and scraps from 2010 to 2050 (Supplementary Fig. 9 and Table 11). Illegal importation of e-waste will shrink and disappear in 2020s. The total importation of scraps was around 21 Mt in 2010, but will dramatically decrease to 1 Mt in 2040. In weight of imported scraps, plastics scrap is leading (verified by Brook et al. studies[38]), followed by steel scrap, copper scrap, and aluminum scrap (Fig. 3e).

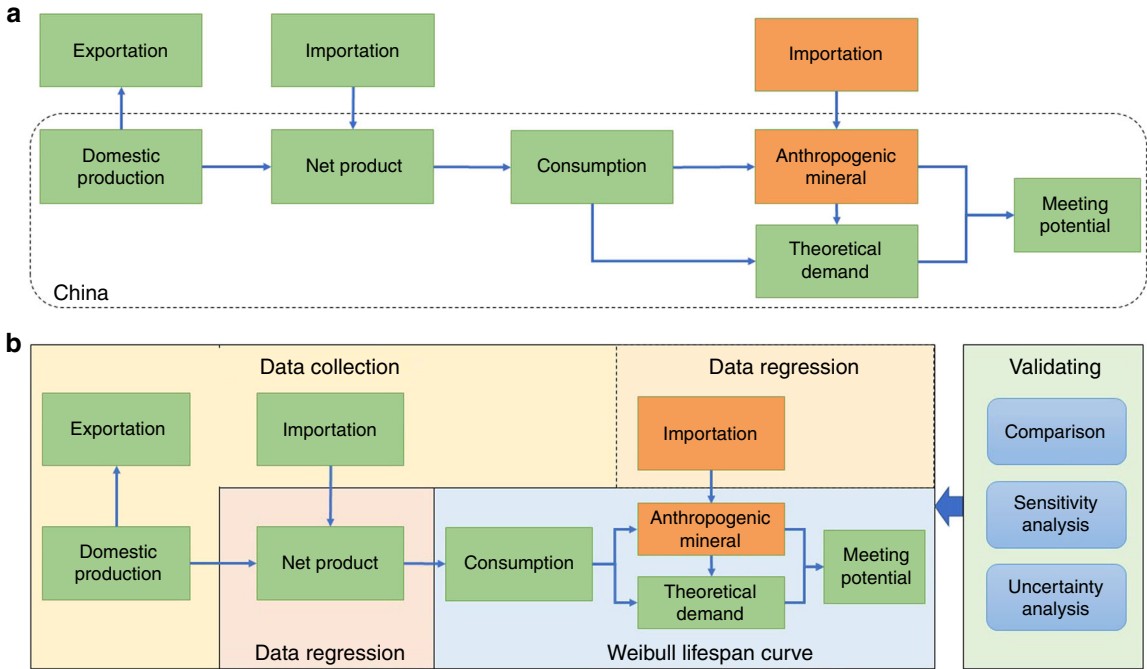

**Fig. 1 The designed framework of research methods in this study. a** Material flow analysis framework for anthropogenic mineral generation and its boundary; Note: green color for product, and orange color for anthropogenic mineral, and dash line indicates the boundary for this study. **b** Four approaches of data collection, data regression, Weibull lifespan curve, and validating methods.

Totally, the weight of the yearly-generated AM in China was about 40 Mt in 2010. Driven by the large expansion of WEEE, ELV, and WWC, total generation weight will reach 71 Mt in 2020, 101 Mt in 2030, and 176 Mt in 2050 (Fig. 2). The average increasing amount in 2010–2050 will be 3.4 Mt per year, and over one half will be provided by ELV, thus ELV may impose more stresses and tensions on government and industry than WEEE and WWC in the future. In addition, importation will shrink when its share in total AM is cut down from 51% in 2010 to 7% in 2025 (Fig. 2).

**Evolution of valuable resources in AM**. When some products reach their EoL, a large quantity of valuable resources is inevitably encased in AM[39], regardless of whether in hibernating stock or not[9,40]. The amount of encased eleven base materials (e.g., Cu, Al, Fe, Co, Pb, Zn, Sn, Mg, plastics, rubber, and glass), five precious metals (e.g., Au, Pd, Ag, Pt, and Rh), two rare metals (e.g., In and W), and five rare earths (including Nd, Dy, La, Y, and Eu) in yearly-generated AM can be easily determined (Supplementary Fig. 11). Almost all mineral resources encased in WEEE, ELV, and WWC have been constantly growing since 2010, and can be expected to continue to rise, at least until 2050.

Totally, for base materials, eight metals, except Pb, maintain the increasing tendency despite a rapid decline of importation. In 2010, the mineral resources of Cu, Al, Fe, Zn, Co, Sn, and Mg were approximately 7.03 Mt, 4.46 Mt, 14.69 Mt, 0.78 kt, 5.82 kt, 0.16 kt, and 46 kt, respectively, but they will rise to 28.52 Mt, 16.35 Mt, 82.63 Mt, 10.65 kt, 23.49 kt, 2.23 kt, and 441 kt on average by 2050, respectively (Fig. 3). The amount of Co and Fe will lead in the increasing rate among all the base materials due to the dramatic and continuing boom of battery and vehicle. The popularity of display substitution and the build-up of vehicles, in particular used with lead-acid battery, have resulted in a peak of 4.3 Mt for Pb generation around 2020, verified by the previous studies[41,42]. Simultaneously, the amount of plastic, glass, and rubber will increase around 55%-fold, 8-fold, and 7.7-fold from 2010 to 2050 (Fig. 3).

The total precious metals are always keeping the growth trend in the year of 2010–2050. The amounts of Au, Pd, Ag, Rh, and Pt were only 89.95, 92.98, 473.84, 2.98, and 6.96 tons, respectively in 2010, but they will grow roughly 15-fold, 19-fold, 14-fold, 8-fold, and 8-fold in 2050 (Fig. 3c). Actually, around global 85% Rh, 50% Pd, and 43% Pt were used in automobile catalyst scraps[43]. The same ascending trend in total rare metals and rare earth remains as the other metals. The amounts were approximately 0.15 kt, 26.75 t, 0.44 kt, and 0.92 kt for In, W, Nd, and other rare earth, respectively, but they will rise to 2.04 Mt, 366 tons, 5.38 kt, and 12.52 kt by 2050, respectively (Fig. 3).

**Economic potential of AM**. From an economic perspective, the large amount of valuable resources accumulated in AM is also enhancing the recycling potential[44]. Despite of the wide range imposed by market price of resource, the average of economic potential from urban mining has been evolving from roughly 74 billion US$ in 2010, to an anticipated 170 billion US$ in 2030 and 428 billion US$ by 2050 (Fig. 4a). During the evolution of recycling potential until 2050, base materials and precious metals have the major economic potential. But the former will drop from 90% in 2010 to around 70% in 2030–2050, and the latter will sharply go up from 4% in 2010 to 15% in 2030–2050 (Supplementary Fig. 12). The remaining recycling potential is predominantly provided by rare metals. Furtherly, the highest-value materials—Cu metal—comprise on average 40%, followed by Au (5–18%), In (4–13%), Al (8–13%), Pd (2–11%), and plastic (5–10%). As a result, Cu, Al, plastic, and Au is currently major recycling targets from AM, but in the future Cu, Au, In, and Al will become the crucial valuable materials (Fig. 4b).

**Meeting potential of AM supply**. The future consumption of product is theoretically equal to net addition to in-use stock,

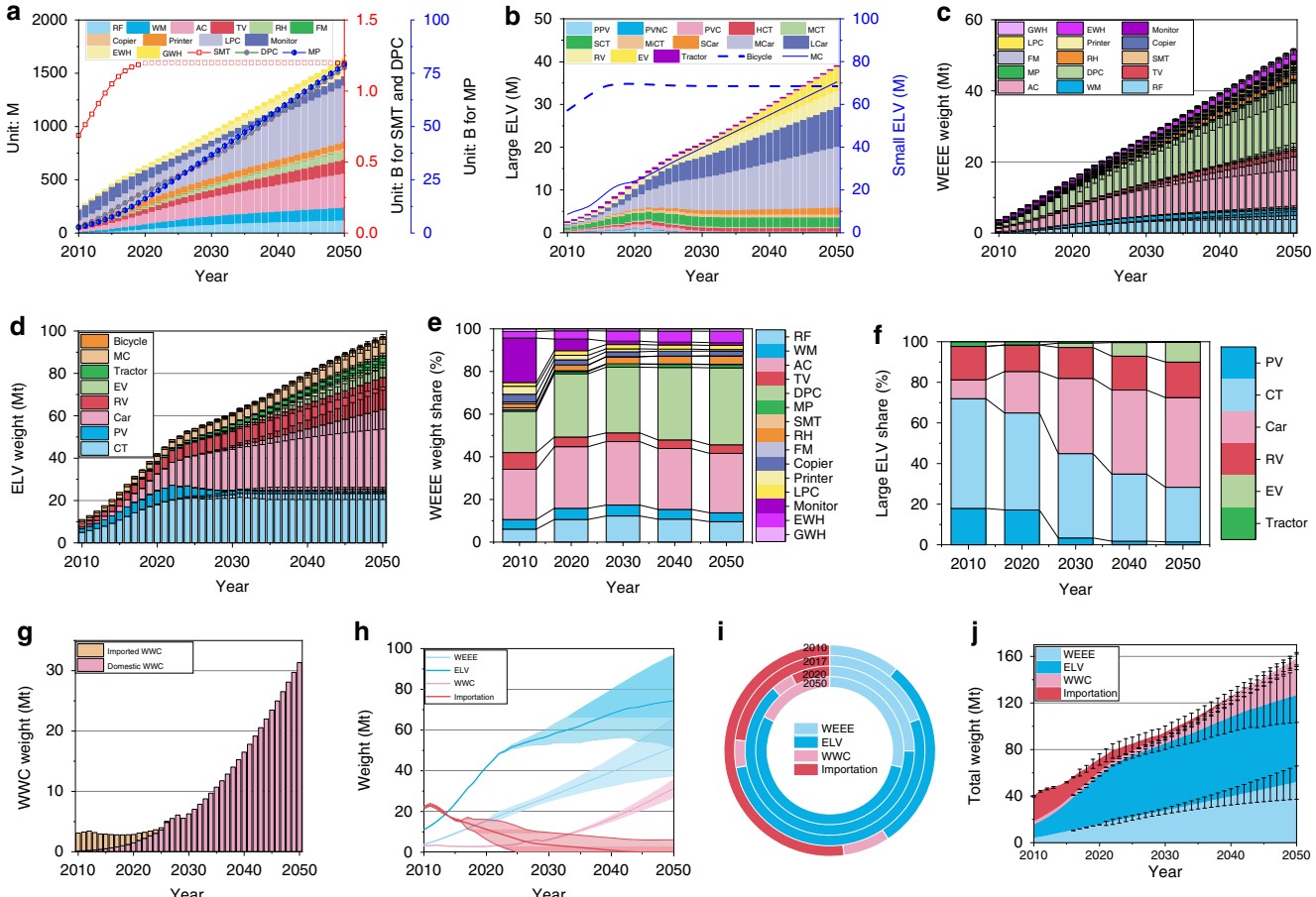

**Fig. 2 Estimation of China's AM from 2010 to 2050. a** WEEE quantity. **b** ELV quantity. **c** WEEE weight. **d** ELV weight. **e** WEEE weight share. **f** ELV weight share. **g** WWC weight. **h** Weight range. **i** Total weight share. **j** Total weight. Note: WEEE consist of AC, WM, RF, TV, DPC, LPC, MP, SMT, RH, printer, copier, monitor, EWH, GWH; Large ELV consists of PPV, NPV, PVC, HCT, MCT, SCT, MiCT, SCar, LCar, RV, EV, and tractor; Bicycle and MC are classified as small ELV; And WEEE, ELV, and WWC make up AM. AC, air conditioner; AM, anthropogenic mineral; CT, cargo truck; DPC, desktop personal computer; ELV, end-of-life vehicle; EV, electric vehicle; EWH, electric water heater; FM, fax machine; GWH, gas water heater; HCT, heavy cargo truck; LCar, large car; LPC, laptop personal computer; M-CRT, CRT monitor used for mainframe; M-FDP, FDP monitor used for mainframe; MC, motorcycle; MCar, medium car; MCT, medium cargo truck; MiCT, mini cargo truck; MP, mobile phone; NPV, New-registration passenger vehicle; PC, personal computer; PCV, passenger civil vehicle; PPV, private passenger vehicle; PV, passenger vehicle; PVC, Private vehicle for civil purpose; RF, refrigerator; RH, range hood; RV, refit vehicle; Scar, small car; SCT, small cargo truck; SMT, single-machine telephone; TV, television; WEEE, waste electrical and electronic equipment; WM, washing machine; WWC, waste wiring and cable.

which is the difference between the demand and the generated AM (Eq. 7 and Supplementary Fig. 1). All the relevant materials are also chosen to uncover the future resource consumption imposed by EEE, vehicle, and wire and cable. Cu, Al, Au, and Pd will maintain the growth until 2030 and afterwards keep stable (Fig. 5), which is attributed to the flourish of wiring and cable and EEE[45]. Pb demand increased from 0.91 Mt in 2000 to 7.75 Mt in 2014[41], but Pb, Fe, Zn, and rubber are quickly decreasing until 2020, and afterwards almost keep the unchanged range while vehicle and EEE are approaching the final saturation; Co will increase and reach the peak in 2030 because the cathode material in lithium-ion battery is gradually substituted from lithium cobalt oxide to lithium nickel cobalt aluminum oxide and lithium iron phosphate in electric vehicle and consumer electronics[46]; Other fifteen materials of future consumption have entered a stabilized phase. The plastic, for instance, will always maintain about 6 Mt in 2010–2050 while the falling importation and the increasing production will mingle together.

We further uncover the supply potential of AM. With the dramatic rise of AM generation and the gradual saturation of

material consumption, the potential supply from AM is becoming possible to overtake the resource consumption of three product groups (Fig. 5). Although we are currently still far from a closed-loop society owing to low recycling rate[47], a rapid advancement is indeed arising for regulation, policy, and technology of circular economy and urban mining. The highly-efficient collection and the cutting-edge recycling will significantly enhance the recycling rate in the future. Thus, if substantial recycling, eighteen materials of AM could meet their demand before 2020, and in 2050 they probably provide over two-fold consumption. The meeting time of Sn and Pd will be approximately 2030 and 2041, respectively. Although Cu, Al, and Co of AM cannot meet their potential consumption by 2050, the disparity gap between their consumption and AM will be greatly reduced in the following decades (Fig. 5w). Consequently, AM supply has a growing potential to meet their future resource consumption.

## Discussion
Deeper discussion is enabled here at three levels of obtained results, potential implications, and recommendation. Firstly, the

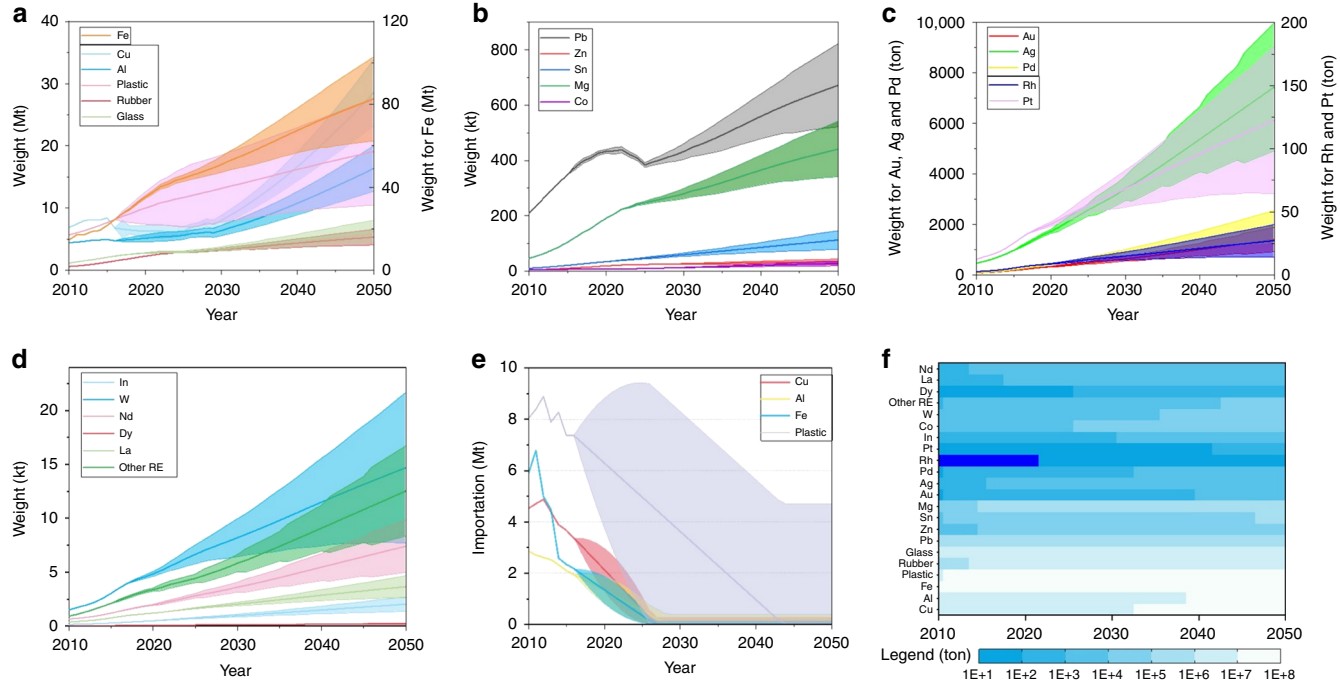

**Fig. 3 Resources weight in yearly-generated AM. a** base material. **b** Pb, Zn, Sn, and Mg materials. **c** precious metals. **d** rare metals and rare earths. **e** imported resources. **f** average resource map. Note: Resources cover the base material (like valuable metal, plastic, and rubber), precious metals, rare metals, and rate earth (RE) elements. The imported resources primarily comprise Cu, Al, Fe, and plastic. The shadowing area indicates the range. For the color map of Rh, deep blue color indicates no data.

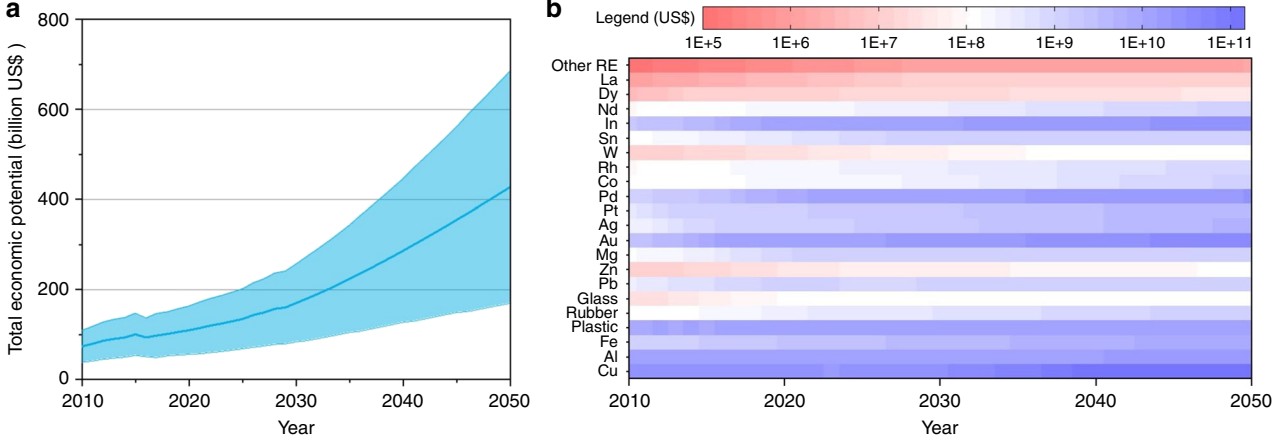

**Fig. 4 Economic potential of AM. a** economic value of total annual AM. **b** economic potential map of materials. Note: the shade of light blue indicates the range of recycling potential. RE, rare earth.

detailed comparison from the previous studies to real-world data was indicated in Supplementary Information Note 4, which can verify the obtained results, and further consolidate the relevant results (Fig. 2). Thus, China has already overtaken the U.S. to become the world's leading producer of e-waste since 2014[24]. MC, car, and EV in quantity will maintain a rapid increment from 2015 to 2050. China provided half of global EV stock with a selling of 1.1 million in 2018[48]. Other vehicles are moving towards saturation (Supplementary Fig. 10). The evolution of WWC weight is subject to longer lifespan of wiring and cable than those of EEE and vehicle. Rapid urbanization and economic growth in 1980s–2010 will impose a significant influence on AM generation after 2025. China has revised the standard of lifespan

for wiring and cable from 20 years to 70 years, which can decrease the obsolescence of wiring and cable. In addition, the Chinese government has issued strict mandates to continue cracking down on smuggling of dangerous trash, medical waste, e-waste, and household garbage (Supplementary Table 10)[37,49]. The importation of AM will fall dramatically leading to eventual cessation of such external sources within a few years (Fig. 2e). Around 99% plastic waste, for instance, has been descended from 7.05 Mt in 2017 to 0.076 Mt in 2018. The solution to plastic waste management is becoming an increasing concern in the European countries, like the UK and France.

Secondly, natural minerals extraction is encountering tremendous challenges. Au, Ag, and In in current global underground

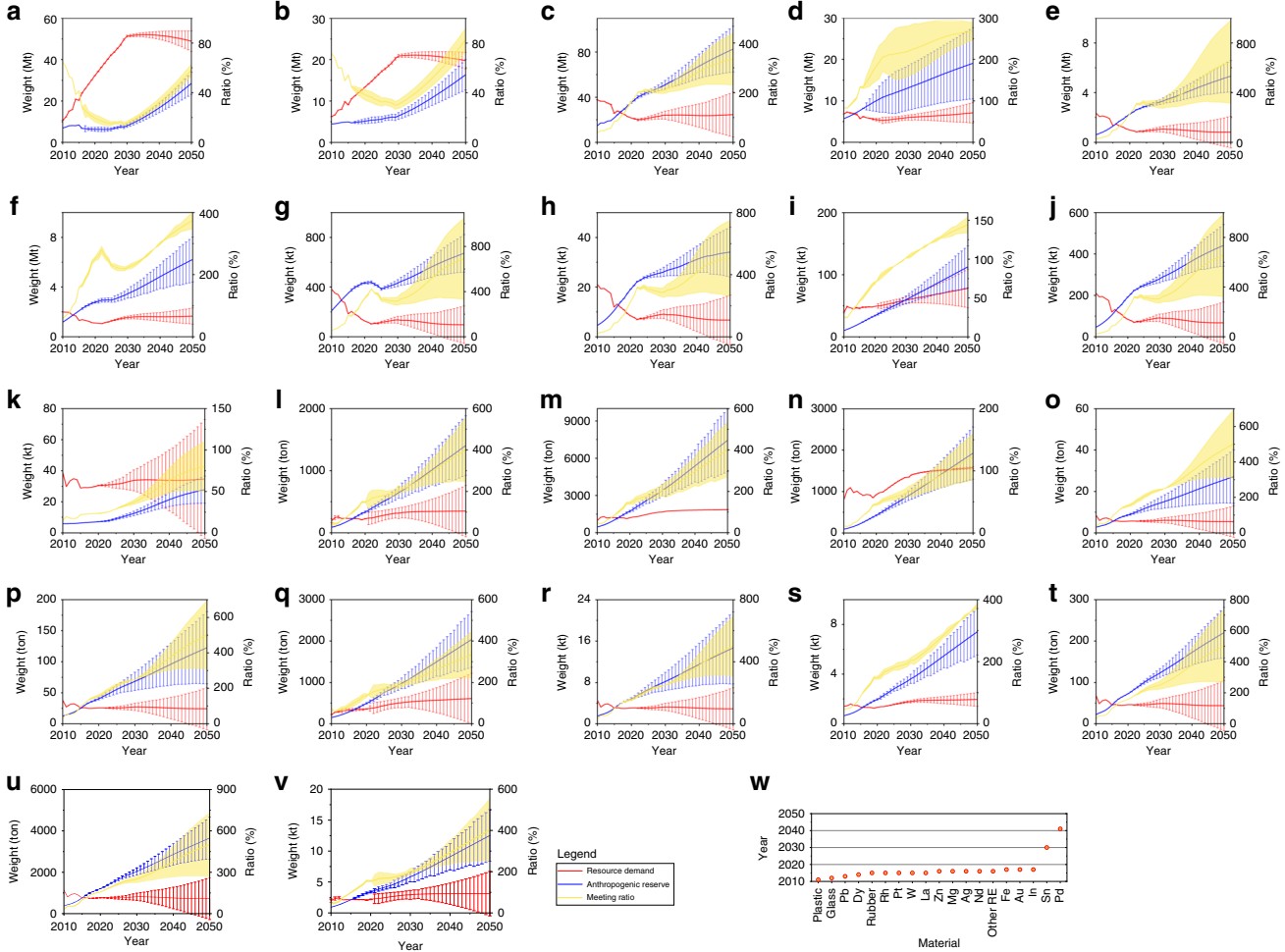

**Fig. 5 Meeting potential of AM for resource consumption of three product groups. a** Cu. **b** Al. **c** Fe. **d** Plastic. **e** Rubber. **f** Glass. **g** Pb. **h** Zn. **i** Sn. **j** Mg. **k** Co. **l** Au. **m** Ag. **n** Pd. **o** Rh. **p** Pt. **q** In. **r** W. **s** Nd. **t** Dy. **u** La. **v** other RE. **w** Meeting time of various materials. Meeting ratio is the value of anthropogenic mineral dividing resource consumption. The shade of light yellowish-brown indicates the range of recycling potential. Rank of various materials is based on the value of meeting time, which is the year while the meeting ratio equals 1.

reserves can only sustain future consumption for 36 years, 9 years, and 4 years[50]. Their exploration and mining will be far harder and more costly owing to the increasing depth of exploration and declining grade[51,52]. Most anthropogenic base metal minerals have a higher grade and purity (as the pure metal or alloy) than the general natural minerals. Eventually, through a circular economy paradigm AMs will play an essential role for supply security and for overall sustainability objectives. China's focus will be shifting from the dependence of virgin mining and waste imports to domestic urban mining of AM.

Lastly, to improve urban mining by circular economy, the following points could be also considered: First, some specific uses of metals (e.g., indium[53], aluminum[54]) were more difficult to fully recycling from AM owing to their high dispersion or potential thermodynamic limit[55]. Further efforts in regulation and more targeted technologies related to the circular economy and zero waste have the potential to ameliorate this situation[56]; Second, investment in research and infrastructure to allow for AM mining can be an effective approach to relieve this resource shortage bottleneck[22,57,58]. Meanwhile, the increase in the amount of AM, as shown by our data, suggests more economic viability in harnessing this reserve; Third, last but not least, the environmental and social risks associated with harnessing AM should be fully considered[59].

With such appropriate policy impetus and precautions, the role of AM in achieving the targets set forth in the United Nations Sustainable Development Goal 12 (responsible production and consumption) are far more likely to be achieved for China and indeed for other rapidly growing economies.

## Methods

**Data collection.** To ascertain the recycling potential of AM, we collected all available material data needed for our supply sources and substantially consolidated the output (Fig. 1), including detailed classification of EEE, vehicle, and wiring and cable (Supplementary Fig. 4); statistics of domestic production for EEE, vehicle, and wiring and cable in China for 1990–2016 (Supplementary Table 1); statistics of importation and exportation of EEE, vehicle, wiring and cable, e-waste, metal scraps, waste plastics for 1990–2016 (Supplementary Table 2); the weight ranges and those data distributions of each WEEE, ELV, and WWC determined from large samples (Supplementary Table 3); the average content of specific resources without any prejudice in a typical AM (Supplementary Tables 4, 5), and uncertainty rates of all the content are estimated as no more than 20% and obey normal distribution[60,61], and the market prices of the resources contained in the AM (Supplementary Table 6).

We collected the global available data and found that the average life spans of private passenger vehicles, government and business vehicles, non-operating buses, heavy duty, medium duty and light duty trucks are 14.5, 13.1, 11.5, 12.8, 10.1, and 8 years, respectively (Supplementary Table 12). Average life spans of taxis, transit buses and non-transit operating buses are 5, 9, and 5.5 years with the pattern of mandatory scrappage[62].

**Material flow analysis framework**. More than ten methods or models have been adopted for AM estimation in previous studies (Their applications and difference can be seen in Supplementary Tables 6, 7). The selection of a particular method depends mainly on data availability and robustness[63,64]. During the process of urban metabolism, products flow into the society (sales), then accumulate in the built environment (stock); when reaching end of life (EoL) after a certain period (lifespan), they flow out as an AM[65]. MFA models quantitatively describe the dynamics, magnitude and interconnection of product sales, stocks and lifespans[66]. Along this flow, China's AM can be sourced both domestically consumed products and imported waste (Fig. 1). Domestic AM yield is attributed to products' manufacturing, exportation, and importation. Therefore, total weight of the AM can be defined by

$$T(x) = D(x) + I'(x) \tag{1}$$

where $T(x)$ is total China's AM by weight (in ton), $x$ is the year; $D(x)$ is domestic AM weight (in ton), and $I'(x)$ is imported AM weight (in ton).

**Data regression**. Any given products as the net production from Eq. 1 was assumed to be both consumed and discarded in China, and its future demand was then determined using a time-step method based on data regression (Fig. 1). Ideally, the inverted-U shape, in particular for flying geese evolving pattern[67], can describe the relationship between resource consumption and economic growth. It is characterized of the rapid growth at the start stage, stable growth at the middle stage, slow growth for a constant, and final decrease at the late stage, and could be expressed by exponential function, linear function, constant value, and linear function, respectively. In addition, the decline of production and consumption commonly occurs while the product is replaced. This principle is substantially applied to align the unavailable data.

**Weibull lifespan curve**. Although products updating and consumer lifestyles can affect the utilization of products, the theoretic lifespan of products relies on the fatigue of the component materials subject to utilization. The Weibull curve is sophisticated to depict the relationship between endurable fatigue and life time (or cycles to failure)[68]. In particular, the Weibull statistical distribution was chosen for this study to model the lifetime of product (Fig. 1). For no regulated-lifespan products like EEEs and bicycle, the probability density function (PDF) of the Weibull distribution is given by Eq. 2. Regarding the regulated-lifespan products like vehicle, and wiring and cable, the regulated lifespan should be considered for a revised Weibull lifespan curve (Eq. 3).

$$f(x) = \begin{cases} \frac{\beta}{\eta}\left(\frac{x}{\eta}\right)^{\beta-1} e^{-(x/\eta)^{\beta}} & x \geq 0 \\ 0 & x < 0 \end{cases} \tag{2}$$

$$f(x) = \begin{cases} 0 & x > L \\ e^{-(x/\eta)^{\beta}} & x = L \\ \frac{\beta}{\eta}\left(\frac{x}{\eta}\right)^{\beta-1} e^{-(x/\eta)^{\beta}} & 0 \leq x < L \\ 0 & x < 0 \end{cases} \tag{3}$$

The cumulative distribution function (CDF) for the Weibull distribution is[69]

$$F(x) = 1 - e^{-(x/\eta)^{\beta}} \tag{4}$$

where $\beta$ is the shape parameter ($\beta > 0$), $\eta$ is the scale parameter ($\eta > 0$), and $L$ is the maximum lifetime of products regulated in China's legislation system (in year) (Supplementary Table 13). EoL units for a particular time $x$ can be mathematically described as reference[70]. Eventually, the AM generation, resource stock, and meeting potential can be determined by the following equations:

$$D(m) = \sum_{i=1}^{31} \int_{x_0}^{m} fi(x) \cdot D_i'(x) \cdot wi\, dx = D(m)_{\text{WEEE}} + D(m)_{\text{ELV}} + D(m)_{\text{WWC}}$$

$$= \sum_{i=1}^{15} \left[ f_i(m-1990) \times D_i'(1990) + f_i(m-1991) \times D_i'(1991) + K + f_i(1) \times D_i'(m-1) \right]$$

$$+ \sum_{i=1}^{15} \left[ f_i(m-1991) \times D_i'(1991) + f_i(m-1992) \times D_i'(1992) + K + f_i(1) \times D_i'(m-1) \right]$$

$$+ \left[ f_i(m-1996) \times D_i'(1996) + f_i(m-1997) \times D_i'(1997) + K + f_i(1) \times D_i'(m-1) \right] \tag{5}$$

$$D'(x) = P(x) - E(x) + I(x) \tag{6}$$

$$C(x) = D'(x) - D(x) \tag{7}$$

$$Dm,j = \sum_{i=1}^{31} \left[ \int_{x_0}^{m} fi(x) \cdot D_i'(x) \cdot wi\, dx \cdot cij \right] \tag{8}$$

$$Cm,j = D_{m,j}' - D_{m,j} \sum_{i=1}^{31} \left\{ [D_i'(m) - D_i(m)] \cdot wi \cdot cij \right\} \tag{9}$$

$$M = \frac{D}{C} \times 100 = \frac{D}{D' - D} \times 100 \tag{10}$$

where $m$ is the assigned year (2010–2050) for the concerned generation; $i$ is the $i$th category of product; 31 is total estimated categories for EEE, vehicle, and wiring and cable; $x_0$ is the initial year of production ($x_0 = 1990$ for EEEs, $x_0 = 1991$ for vehicle, and $x_0 = 1996$ for wiring and cable); $D'(x)$ is the demand quantity of product flow from production, exportation, and importation (in million or ton); $C(x)$ is the theoretical consumption of product (in million or ton); $w_i$ is the weight of $i$th category of each product (in kg or ton); $P(x)$ is the domestic production quantity (in million); $E(x)$ is the exportation of product (in million); $I(x)$ is the importation of product (in million); $j$ is the $j$th resource category; $c_{ij}$ is the content of the $j$th resource in the $i$th category of products; $W_{m,j}$ is the total weight of $j$th resource in the yearly-generated AM at the year of $m$ (in ton), which indicates the supply potential of AM; $C_{m,j}$ is the total weight of $j$th resource in the consumption at the year of $m$ (in ton), and $M$ is defined as the meeting ratio of a certain resource supply from AM to the resource consumption from the three product groups (%).

**Sensitivity analysis**. Demand prediction based on data regression, market price of materials, and importation of AM will be evaluated using sensitivity analysis (Fig. 1) because we only know their errors and ranges (Supplementary Tables 5, 11). An incremental error for the future demand of EEEs, vehicle, and wiring & cable is shown in Supplementary Table 14. Correspondingly, the sensitivity using the feasible errors or ranges is indicated for the weight of AM and their materials. Importation is consisted of the legal importation for metals scrap (e.g., Cu, Al, and Fe) and plastic scrap, and the illegal importation for toxic waste (e.g., e-waste). Those ranges will affect the uncertainty of total AM. High but known importation in recent years means high influence on total AM. However, the low range or error of importation in recent years means that it is not sensitive for share of importation. With the error increasing and urban mining growth, the biggest uncertainty occurred in 2023 when importation share ranges from 4% to 19%. Later, rise of domestic AM and decline of importation result in that importation's biggest sensitivity imposed on total AM will drop to 10% in 2030, and less than 4% in 2050 (Supplementary Fig. 5a).

**Uncertainty analysis**. Both the method and the input data will affect the uncertainty of estimated results. The method, in particular from data regression, will be identified with the feasible error. The data including weight and average content will be assessed with uncertainty analysis (Fig. 1). Here, a Monte Carlo simulation ($10^5$ iterations) was conducted to obtain final estimates of flows and their uncertainties in this study. Based on the relevant collected data, and those distributions illustrated in Supplementary Tables 3, 4, 13, uncertainties of total AM weight and yearly-added resource stock (e.g., Cu and Al) in 2017 and 2030, for instance, are performed and presented in Supplementary Fig. 5. Similarly, the resources stock from Monte Carlo simulation can also cover the forecasting results at the maximum probability interval.

**Reporting summary**. Further information on research design is available in the Nature Research Reporting Summary linked to this article.

## Data availability
The authors declare that the data supporting the findings of this study are available within the paper (and its supplementary information files).

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

## Acknowledgements

X.Z. acknowledges funding by National Key Technology R&D Program of China (2018YFC1900101 and 2019YFC1903711).

## Author contributions

X.L.Z. collected data and wrote the paper; S.H.A. refined the narrative and policy analysis; J.P.T. contributed the analytical design; X.L.Z. and J.H.L. supervised this study.

## Competing interests

The authors declare no competing interests.
