## [Peer Review File · Nature Communications]

Editorial Note: In their review of the first version of this manuscript, reviewer #2 added his/her comments to the manuscript file. These comments have not been copied into this Peer Review File.

Reviewers' comments:

Reviewer #1 (Remarks to the Author):

This paper develops a material flow model to estimate the accumulation of key metals and materials in Chinese waste flows and then compares these to estimates of materials demand. The topics covered by this paper are extremely important and would influence thinking in the field. The scope of the analysis described here is novel.

First of all, I would like to congratulate the authors on the bold scope for this important work. The problem you have tackled is important and the results that derive from your analyses are interesting. I would suggest that the authors address the following issues to create an excellent manuscript.

1) The English in this manuscript is challenging throughout. I make this comment in humble recognition that I myself cannot write a single word in Chinese or any other language for that matter. Nevertheless, English is the language required by this journal and as such this will need to be addressed. An example of this include

a. (lines 36-38) "Historically, the called "Four Big Items" is the fashionable aspiration for products (see Supplementary Figs. 1 and 2 in Supplementary information), and its rapid evolution since 1970s has revealed the technological progress and the economic growth."

This sentence is structured awkwardly and as such does not contribute to the narrative.

2) Methods: My most fundamental concern with the work as it is framed currently is that it seems to imply conclusions that are more wide-ranging than the analysis would support. As I understand the analysis, it includes the material flows from three product categories: end-of-life electronics, end-of-life vehicles, and waste wire & cable. These are then discussed as if they represent the only (or at least the major) relevant sources and consumers of the materials studied. There are clearly other relevant classes of products for both overall materials flows (probably the most notable one being buildings and other forms of construction) and for specific materials that are analyzed in this paper (e.g., wind turbine installations for REEs).

I would contrast this scope of analysis with the current title of the paper "Anthropogenic mineral supply has potential to meet Chinese resource demand by 2050" and the second to last sentence of the abstract "All the obtained results demonstrate that anthropogenic material supply can become the dominant source to meet China's mineral demand around 2050." Both of these appear to be all-encompassing conclusions, while the actual analysis is clearly bounded.

I recognize that comprehensive scope in an analysis like this is not possible and that interesting conclusions are possible from the scope that is presented. As such, I would recommend

a. Explain why these product flows (EEE, ELV, and WWC – I will call them focal products) were selected for analysis. Probably ready access to data, but this should be made explicit.

b. Select materials (I will call them focal materials) that are interesting to you and where you can make a reasonable case that flows associated with the focal products represent a majority of the flow of the focal materials. A clear example of this would be Pd. Auto catalysts dominate Pd uses. Therefore, modeling Pd flows from ELVs provides good insight into balance between Pd demand and its secondary supply. A clear example of a concerning example would be Fe. I don't believe that you are tracking

supply or demand from building construction. Without this, it seems difficult to comment on the annual net addition to in-use stock for Fe.

c. Comment on flows NOT associated with the focal products. I fully recognize that modeling other product flows is not likely to be possible. Nevertheless, you should try to comment on what fraction of total material flow is associated with the focal products and be explicit that you are assuming that patterns in non-focal products will be similar to those in the focal products. Finally, you should flag any product/material flow patterns which would likely not mimic the focal product. An example here might be dental applications of Pd which might be considered as dissipative.

d. Focus your analysis and conclusion on the focal materials. You should certainly comment on other materials, but the materials for which you can reach the strongest conclusions are those where you are modeling the majority of their flow.

3) METHODS: I believe that the reader would benefit from a more comprehensive statement of the equations that define your model. For example, in the results section you state "the annual net addition to in-use stock can be defined as the difference between annual net production and annual domestic anthropogenic mineral generation". There should be formulae (connected to the existing set of formulae) that map the currently defined variables to these final metrics.

a. The relationship between equation 4 ($F(x)$) which would be unitless and range between 0 and 1) and equation 6 within which $F(x)$ is listed as having units of tons is unclear.

4) METHODS: It would be easier to follow your method if you reversed the order of the sections in the Methods section. "Estimating anthropogenic mineral generation" first and probably move the "Data" subsection to the end of the Methods section. By doing this, you can use the symbols defined in equations 1-6 to label each of the types of data described in the Data subsection.

5) RESULTS: In the section "Dynamic transfer map of valuable resources." It is unclear why you discuss net addition to in-use stock for only seven of the 21 materials that you were tracking. Presumably this is because of your conclusion that they are the dominant value generators. If this is so, please make this explicit.

6) RESULTS: In the presentation of your results, it would be valuable to include a plot that shows the evolution of both the supply (annual domestic anthropogenic mineral generation) and the demand (annual net production) for at least one focal material, hopefully, putting these on the same plot. This would help readers to better visualize the definition of the annual net addition to in-use stock as the gap between these two quantities. I would also suggest calling out the definition of annual net addition in a formula. A formula in this context is needed not because the relationship is complex, but rather because it helps to make the definition of this key result stand out in the manuscript.

7) RESULTS: As noted in the title and the abstract, you draw a conclusion that anthropogenic minerals can meet resource demand by 2050. Presumably, this is based on the analysis presented in Figure 4. Because this is such an important conclusion, this paper would be greatly improved if you spent more time discussing these figures and how someone can map these results to the conclusion about meeting resource demand.

Just to explain what I mean here, consider these points. In the way that this section is written up currently, it appears that Annual Net Addition to In-Use Stock is the ultimate metric used to support your conclusion. Presumably as that quantity goes to zero it implies that anthropogenic mineral

generation is occurring at a rate comparable to the rate of production. If one looks naively at figure 4a, one might conclude that anthropogenic minerals are only going to be sufficient for Pb because it is the only one for which the result approaches zero.

Randolph Kirchain

Reviewer #2 (Remarks to the Author):

Dear editor and authors,

I have now had time to read and consider this manuscript in detail. In this document, I am providing some general comments, with more specific comments in the PDF of the article and supplement which I upload with this review.

With this contribution, the authors set out on an ambitious mission to not only assess the material flows from various waste products in China in the future (up to 2050), but they also purport to compare the supply potential for different metals and other materials from these waste streams to the total raw materials demand of China. They conclude in the abstract, with their own words that:

"All the obtained results demonstrate that anthropogenic material supply can become the dominant source to meet China's mineral demand around 2050." (lines 12 – 14)

While the general subject of this manuscript is certainly interesting, it is not suitable for publication in Nature Communications in its current state. Not only is the quality of the presentation considerably below the standard of the journal, but there are also substantial flaws in the analysis which mean that the conclusion I quoted from the abstract above is certainly not supported by the facts and analysis presented by the authors. I explain in more detail why this is below.

Style and quality of presentation

While I understand that English is not the first language of the first author, I found this manuscript extremely difficult to read on account of the quality of its descriptions. Not only does it use excessive phraseology which renders some sentences virtually unintelligible, but there are also logical gaps and wrongly used terminology that make the reasoning of the authors hard to follow (e.g. lines 41 to 43). As another example of this I would like to use the term "stocks" which is generally taken mean a cumulative quantity. However, the authors use it to mean annual quantities e.g. of waste generated. I only realized this towards the end of the manuscript and got very confused when trying to make sense of figures 2 and 3. I could quote more examples. I am actually surprised that such large deficiencies have not been corrected by the second author, whose English language skills I would have expected to be much better, given that he has held teaching positions at academic institutions in both Australia and the US.

Another problem is that figures are not well designed and labelled. For instance, Fig. 2a – e and 2f essentially represent the same information in different ways. It would have been better to just use the more compact of these representations. This will also reduce print space. As it is, there is unnecessary redundancy that does not provide any significant gain in information.

Furthermore, the text contains several statements that are simply factually wrong, e.g. that China is largely reliant on imported raw materials (lines 40 and 280/281). Quick reference to the USGS Mineral

Commodity Summaries will show that in fact, China is the main producer of many primary metals (Zn, Pb, Ga, Ge, In, Sb, REEs etc.). Another such statement is that recycling is generally cheaper and “better” source for ALL metals than mining (lines 277 / 278). This is similarly untrue. It depends very much on the metal in question. See Reuter et al. (2011) and Allwood (2014). For instance, Al cannot currently be recycled to yield the same material quality as primary Al. Indium is too greatly dispersed in its major applications to warrant recycling. Extraction from primary sources (zinc ores) is cheaper.

Methodology and soundness of analysis

While none of the deficiencies listed above would never have led me to reject this manuscript had the authors’ analysis been sound, this does, unfortunately, also contain some major deficiencies as indicated above. I will start with the most important problem. The authors make a statement about the potential importance of secondary resources to supply the needs of China in 2050. It is also implicit in the manuscript’s title. I would have expected such a strong statement to be supported by a similarly strong analysis of BOTH the expected quantity of waste materials and metals contained within them (as well as estimates of recoverable quantities) AND the expected demand from the Chinese economy. However, the authors only provide a very simplistic analysis of the expected amounts of waste materials, and do not at all consider the demand side anywhere. It is never mentioned. This fact alone invalidates both the title and the supposed major conclusion of this manuscript.

Nevertheless, a proper quantitative analysis of waste streams and their probable evolution could still have been valuable in its own regard. However, this is not what the authors purportedly set out to represent, nor is their current analysis without flaws. My major points of criticism in this regard center on: 1) extremely simplistic forecasting of waste flows, and 2) insufficient analysis of uncertainties.

Forecasting. To assess the amounts of materials ending up as waste, the authors use a combination of production statistics for the products they investigated (from 2003 – 2013) and their extrapolation into the future in combination with a probabilistic description of likely product lifetimes. My problem here mainly concerns the extrapolation into the future. Based on mostly linear but also sometimes exponential fits (not clear, why two methods are used, and what the justification is in specific cases) to the 11 years or so for which they have data, they simply extrapolate 35 years into the future (!). This is extremely naïve. Not only is not clear that linear growth is a good representation of the likely future development for different product categories, particularly those showing a decrease in production volume over the time-span considered (they would for several products end up negative). But it is also questionable that these trends are robust over a time span that is more than THREE TIMES the range spanned by the input data. Which brings me to the next point: uncertainty analysis.

Uncertainty analysis. While the authors supposedly include a very crude uncertainty analysis, they show corresponding confidence intervals in none of their figures. I would have expected to see these, using the standard 95% reporting method. Furthermore, the authors only consider one major source of uncertainty – the composition of waste materials. However, they ignore an even more substantial one: the very large uncertainties that will result from their forecasting methods. All of their fits to production data will have uncertainties. The further these trends are extrapolated into the future, the larger these uncertainties become. Given the large degree of extrapolation, it is likely that by 2050 these uncertainties will be so large that definitive statements on available amounts to within a factor 2 or less will be nearly impossible. This needs to be considered.

Results and conclusions

Given the many reservations I have already voiced above, I do not think it is sensible to comment in

detail on these sections. They will at any rate require a complete rewriting to account for the changes made elsewhere in the manuscript to address my recommendations (below).

Suggestions for improvement

Following on from my comments above, I have two major suggestions for improvement. First, I suggest the authors rethink the purpose of this publication. With their current analysis, neither title nor major conclusion are justified. If they wish to make these conclusions, they also need to include an assessment of the demand side for China up to 2050. It may be easier to just focus on supply.

Second, I suggest the authors incorporate the fitting uncertainty of their linear (and other) regression analysis into their uncertainty assessment. I encourage them to try out different functional forms. I expect they will be surprised at how large these uncertainties become decades into the future. Third, I suggest the authors give the revised text that incorporates the changes suggested above a complete overhaul to improve the quality and style of their language, and ensure a concise and coherent presentation, ideally with the help of a native speaker or a professional language editing service. This should ensure that the manuscript meets the quality requirements of the journal and is easier on readers.

References cited

Allwood JM (2014) Squaring the circular economy: The role of recycling within a hierarchy of material management strategies. In: Worrell E, Reuter M (Eds) Handbook of Recycling, Elsevier, Amsterdam-New York-London.

Reuter MA (2011) Limits of design for recycling and "sustainability": A review. Waste Biomass Valor 2:183-208.

Reviewer #2

This paper develops a material flow model to estimate the accumulation of key metals and materials in Chinese waste flows and then compares these to estimates of materials demand. The topics covered by this paper are extremely important and would influence thinking in the field. The scope of the analysis described here is novel.

First of all, I would like to congratulate the authors on the bold scope for this important work. The problem you have tackled is important and the results that derive from your analyses are interesting. I would suggest that the authors address the following issues to create an excellent manuscript.

Response: Thanks for your valuable comments and careful suggestions. All the suggestions have been fully considered to improve the article.

1) The English in this manuscript is challenging throughout. I make this comment in humble recognition that I myself cannot write a single word in Chinese or any other language for that matter. Nevertheless, English is the language required by this journal and as such this will need to be addressed. An example of this include

a. (lines 36-38) “Historically, the called “Four Big Items” is the fashionable aspiration for products (see Supplementary Figs. 1 and 2 in Supplementary information), and its rapid evolution since 1970s has revealed the technological progress and the economic growth.”

This sentence is structured awkwardly and as such does not contribute to the narrative.

Response: Thanks. We rewrote some sentences again and found the native English speaker to polish the language. Please check up the revised article.

3. 2) Methods: My most fundamental concern with the work as it is framed currently is that it seems to imply conclusions that are more wide-ranging than the analysis would support. As I understand the analysis, it includes the material flows from three product categories: end-of-life electronics, end-of-life vehicles, and waste wire & cable. These are then discussed as if they represent the only (or at least the major) relevant sources and consumers of the materials studied. There are clearly other relevant classes of products for both overall materials flows (probably the most notable one being buildings and other forms of construction) and for specific materials that are analyzed in this paper (e.g., wind turbine installations for REEs).

Response: Many thanks for your nice concern on the scope. High-tech products are not only the core

driver of economic growth, but also the major carrier of resource consumption and environmental pollution (direct or indirect). Their rapid popularity in recent decades and hopefully in the coming years are depleting the primary mineral resources, in particular for the strategic (or critical raw) materials. They are finally sinking in anthropogenic mineral (or urban mineral), which is regarded as one of the fastest-growing solid waste stream. In China, the government body, National Development and Reform Commission in 2010 proposed a distinct concept of anthropogenic mineral defined as the recycled iron, non-ferrous metals, precious metals, plastic, or rubber material lain in three large groups of waste products such as electrical and electronic equipment (EEE), vehicles, and wires and cables, generated at the duration of industrialization and urbanization. Although there are somewhat different types of anthropogenic mineral among the world, the three types, WEEE, ELV, and WWC, are the main and common core of anthropogenic mineral as the most typical resource-contained waste. Other waste like construction & demotion waste, municipal waste, and landfilled waste will be not considered. Basically, solid waste can be classified as the resource-contained waste (e.g., WEEE, ELV, metal scrap) and the environmental-risk waste (e.g., fly ash and municipal solid waste). This study only considers the resource-contained waste, which can provide the increasing potential of resource supply.

... However, the shortening useful life expectancy of the product, driven by rapid innovation, miniaturization and affordability, and an increasingly anthropogenic metabolism have led to a major increase in the accumulation of waste products, which could potentially be a secondary resource (Terminologies definition and boundaries are available in Supplementary Information Section 1).^{3, 5} The scope of such anthropogenic sources is strictly regulated by the Chinese government and comprises recycled iron, non-ferrous metals, precious metals, plastic, or rubber material embedded in three groups of waste products such as electrical and electronic equipment (EEE, all the abbreviations and acronyms are provided at Supplementary Information Section 2), vehicles, and wires and cables, generated during the industrialization and urbanization.^{6, 7} These three groups of waste products are identified as the core anthropogenic mineral types not only in China, but also in many industrial nations.^{8, 9, 10, 11} Intrinsically, anthropogenic minerals are secondary resources that have already been crafted with specific human activities in their design and

thus comprise an optimal mix of functional minerals in their supply chain.⁷ Critical raw materials have also been sinking into this supply source, which is accelerating the depletion of virginal resources and frustrating the sustainable development of the manufacturing industry.^{12, 13, 14, 15}

4. I would contrast this scope of analysis with the current title of the paper “Anthropogenic mineral supply has potential to meet Chinese resource demand by 2050” and the second to last sentence of the abstract “All the obtained results demonstrate that anthropogenic material supply can become the dominant source to meet China's mineral demand around 2050.” Both of these appear to be all-encompassing conclusions, while the actual analysis is clearly bounded.

Response: We added new session to examine the meeting potential for demand. Meanwhile, we polished carefully the logic of text and paragraph in Introduction and Discussion session to well support the title and abstract. Please check up. Here are some typical revisions.

...

EEEs and vehicles are the most fashionable aspiration of assets in Chinese households, which are the hallmark of the once-called “Four Big Items” that consumers are aspiring towards (see Supplementary Figs. 3 and 4). Their rapid evolution and popularity since 1970s could reveal a dramatic rise in waste generation and resource consumption. The consumption of a couple of mineral resources has gone through multiple increases²⁰, resulting in a shortage of some important strategic resources and a growth of external dependence.^{21, 22, 23} To meet the future resource demand, resources mining from anthropogenic mineral has become a world concern at the context of circular economy.

...

Therefore, we seek to uncover the mechanism of urban metabolism for e-waste, ELV, and WWC, comprehensively discover the recycling potential of anthropogenic mineral, and demonstrate their potential to meet future resource demand. To complete this study, four pathways are employed: data collection for the consumption, importation, exportation, and material composition of anthropogenic supply; method development based on material flow analysis and urban metabolism; demand estimation for anthropogenic mineral quantity, and resultant economic potential; and validation of results using comparison with previous studies or reported data, sensitivity analysis, and uncertainty analysis (Supplementary Figs. 5 & 7).

...

Meeting potential of anthropogenic mineral supply for resource demand. *The future demand of product is theoretically equal to net addition to in-use stock, which is the difference between the consumption and the generated anthropogenic mineral (Eq. 7 and Supplementary Fig. 2). All the relevant materials are also chosen to uncover the future resource demand. Cu, Al, Au, and Pd will maintain the growth until 2030 and afterwards reach the constant (Fig. 4), which is attributed to the flourish of wiring and cable and EEE⁷⁰. Pb demand increased from 0.91Mt in 2000 to 7.75 Mt in 2014⁶⁶, but Pb, Fe, Zn, and rubber are quickly decreasing until 2020, and afterwards almost keep the unchanged range while vehicle and EEE are approaching the final saturation; Co will increase and reach the peak in 2030 because the cathode material in lithium-ion battery is gradually substituted from lithium cobalt oxide to lithium nickel cobalt aluminum oxide and lithium iron phosphate in electric vehicle and consumer electronics⁷¹; Other fifteen materials of future demand have entered a stabilized phase. The plastic, for instance, will always maintain about 6 Mt in 2010-2050 while the falling importation and the increasing production will mingle together.*

...

Globally, Au, Ag, and In in remaining underground reserve can only sustain demand for 36 years, 9 years, and 4 years at half of U.S. per capita consumption rates.⁷³ Their exploration and mining will be far harder and more costly owing to the increasing depth of exploration and the descending grade.^{74, 75} Most anthropogenic base metal minerals have the higher grade and purity (as the pure metal or alloy) than the general natural minerals. Eventually, anthropogenic minerals will play an essential role for supply security and for overall sustainability objectives. Although some specific uses of metals require specific properties that are more difficult to obtain from secondary source metals⁷⁶, refinement and metallurgical processes are constantly being improved for resource recovery.⁷⁷ Investing in research and infrastructure to allow for anthropogenic mineral recovery can be an effective approach to relieving this resource shortage bottleneck,^{22, 78, 79} while China relies on imports for many primary minerals⁸⁰. Meanwhile, the increase in the amount of anthropogenic mineral, as shown by our data, suggests more economic viability in harnessing this reserve. At the same time the environmental and social risks associated with harnessing anthropogenic minerals should also be considered alongside those risks for conventional primary mining.

5. I recognize that comprehensive scope in an analysis like this is not possible and that interesting conclusions are possible from the scope that is presented. As such, I would recommend

- a. Explain why these product flows (EEE, ELV, and WWC – I will call them focal products) were selected for analysis. Probably ready access to data, but this should be made explicit.
- b. Select materials (I will call them focal materials) that are interesting to you and where you can make a reasonable case that flows associated with the focal products represent a majority of the flow of the focal materials. A clear example of this would be Pd. Auto catalysts dominate Pd uses. Therefore, modeling Pd flows from ELVs provides good insight into balance between Pd demand and its secondary supply. A clear example of a concerning example would be Fe. I don't believe that you are tracking supply or demand from building construction. Without this, it seems difficult to comment on the annual net addition to in-use stock for Fe.
- c. Comment on flows NOT associated with the focal products. I fully recognize that modeling other product flows is not likely to be possible. Nevertheless, you should try to comment on what fraction of total material flow is associated with the focal products and be explicit that you are assuming that patterns in non-focal products will be similar to those in the focal products. Finally, you should flag any product/material flow patterns which would likely not mimic the focal product. An example here might be dental applications of Pd which might be considered as dissipative.
- d. Focus your analysis and conclusion on the focal materials. You should certainly comment on other materials, but the materials for which you can reach the strongest conclusions are those where you are modeling the majority of their flow.

Response: Many thanks for your nice understanding and recommendations.

- a. This reason has been addressed above from the regulated concept of anthropogenic mineral and given in detail in the introduction session.
- b. We try to draw all the relevant materials encapsulated in anthropogenic mineral, including the basic materials (like Cu, Fe, and Plastic), precious metals (like Au, Ag, and Pd), and strategic materials (like In, Ga, and rare earth). The highly valuable and critical materials are paid a more concern in the high-tech products. Previous studies indicate that Au, Pd, and In are the most typical to maintain the balance between the secondary supply and their industry demands. Regarding the

in-use stock, only the three large groups and thirty products are concerned, not covering the building construction. Therefore, here Fe quantity of in-use stock is not so higher.

c. Please find the improved documents.

d. Thanks. Main materials like copper, precious metals, and rare metals are focused. We expanded all the relevant materials to support the conclusion.

6. 3) METHODS: I believe that the reader would benefit from a more comprehensive statement of the equations that define your model. For example, in the results section you state “the annual net addition to in-use stock can be defined as the difference between annual net production and annual domestic anthropogenic mineral generation”. There should be formulae (connected to the existing set of formulae) that map the currently defined variables to these final metrics.

a. The relationship between equation 4 (F(x) which would be unitless and range between 0 and 1) and equation 6 within which F(x) is listed as having units of tons is unclear.

Response: revised carefully, thanks!

...Therefore, total weight of the anthropogenic mineral can be defined by

$$T(x) = D(x) + I'(x) \quad (1)$$

Where $T(x)$ is total China's anthropogenic mineral by weight (in ton, 1 ton = 1000kg), x is the year; $D(x)$ is domestic anthropogenic mineral weight (in ton), and $I'(x)$ is imported anthropogenic mineral weight (in ton).

...

$$D(m) = \sum_{i=1}^n \int_{x_0}^m f_i(x) \cdot C_i(x) \cdot w_i dx$$

$$\begin{aligned} &= \sum_{i=1}^{15} [f_i(m-1990) \times C_i(1990) + f_i(m-1991) \times C_i(1991) + K + f_i(1) \times C_i(m-1)] \\ &+ \sum_{i=1}^{15} [f_i(m-1991) \times C_i(1991) + f_i(m-1992) \times C_i(1992) + K + f_i(1) \times C_i(m-1)] \quad (5) \\ &+ [f_i(m-1996) \times C_i(1996) + f_i(m-1997) \times C_i(1997) + K + f_i(1) \times C_i(m-1)] \end{aligned}$$

$$C(x) = P(x) - E(x) + I(x) \quad (6)$$

$$D'(x) = C(x) - D(x) \quad (7)$$

$$W_{m,j} = \sum_{i=1}^{31} \left[\int_{x_0}^m f_i(x) \cdot C_i(x) \cdot w_i dx \cdot c_{ij} \right] \quad (8)$$

$$S_{m,j} = \sum_{i=1}^{31} \{ [C_i(m) - D_i(m)] \cdot w_i \cdot c_{ij} \} \quad (9)$$

$$M = W / S \times 100 \quad (10)$$

where m is the assigned year (2010-2050) for the concerned generation; i is the i^{th} category of product; 31 is total estimated categories for EEE, vehicle, and wiring & cable; x_0 is the initial year of production ($x_0=1990$ for EEEs, $x_0=1991$ for vehicle, and $x_0=1996$ for wiring & cable); $C(x)$ is the consumption quantity of product flow from production, exportation, and importation (in million or ton); $D'(x)$ is the theoretical demand of product (in million or ton); w_i is the weight of i^{th} category of product (in kg or ton); $P(x)$ is the domestic production quantity (in million); $E(x)$ is the exportation of product (in million); $I(x)$ is the importation of product (in million); j is the j^{th} resource category; c_{ij} is the content of the j^{th} resource in the i^{th} category of products; $W_{m,j}$ is the total weight of j^{th} resource in the yearly-generated anthropogenic mineral at the year of m (in ton), which indicates the supply potential of anthropogenic mineral; $S_{m,j}$ is the total weight of j^{th} resource in the yearly-added consumption at the year of m (in ton), which indicates the theoretical demand of resource; and M is defined as the resource supply meeting ratio of anthropogenic mineral for the demand (%).

4) METHODS: It would be easier to follow your method if you reversed the order of the sections in the Methods section. “Estimating anthropogenic mineral generation” first and probably move the “Data” subsection to the end of the Methods section. By doing this, you can use the symbols defined in equations 1-6 to label each of the types of data described in the Data subsection.

Response: thanks for your suggestions. The method choosing is basically relied upon the collected data. Supplementary Table 8 demonstrates previous applications of various methods for predicting anthropogenic mineral generation subject to the data availability. So it will be better to set data first and method following. But we will use the symbols by the article to label the data and results in Supplementary information. It will screw the whole article for easy reading and understanding.

Contents

Supplementary Section 1: Terminologies definition and boundary	S1
Supplementary Fig. 1 Boundary of anthropogenic mineral, natural mineral, primary resources, secondary resources, solid waste, new scrap, and old scrap	S1
Supplementary Fig. 2 Relationship among the production, consumption, and theoretical demand	S2
Supplementary Section 2: Main abbreviations and acronyms.....	S3
Supplementary Fig. 3 Evolution of “Four Big Items” from 1970s to 2000s	S4
Supplementary Fig. 4 Evolution of television since 1960s.....	S5
Supplementary Fig. 5 The concise diagram of research route for this study.....	S6
Supplementary Fig. 6 Classification of EEE, vehicle, and wiring & cable in China.....	S7
Supplementary Fig. 7 The designed framework of research methods.....	S8
Supplementary Fig. 8 Sensitivity analysis and uncertainty analysis with 95% confidence intervals	S9
Supplementary Section 3: Available data pre-mining for estimation.....	S10
Supplementary Fig. 9 Probability density function of Weibull distribution.....	S10
Supplementary Fig. 10 Data regression of net production of EEEs for $R(x)$	S11
Supplementary Fig. 11 Data regression of net production of vehicle for $R(x)$	S13
Supplementary Fig. 12 Linear regression of imported scraps for $I'(x)$	S16
Supplementary Fig. 13A Estimation of China’s anthropogenic mineral from 2010 to 2050: Average value for $D(x)$	S17
Supplementary Fig. 13B Estimation of China’s anthropogenic mineral from 2010 to 2050: Range for $D(x)$ (Mt).....	S18
Supplementary Fig. 14 Resources weight in yearly-generated three anthropogenic minerals for S_i	S21
Supplementary Fig. 15 Relative economic shares of materials in total anthropogenic mineral	S22
Supplementary Section 4: Comparison of this study to others studies.....	S23
Supplementary Fig. 16 The comparison of typical anthropogenic mineral estimation.....	S23
Supplementary Fig. 17 Validation for all the registered vehicles quantity	S24
Supplementary Table 1 $R(x)$: Statistics of domestic production for EEE, vehicle, and wiring & cable in China	S25
Supplementary Table 2 $I(x)$ and $E(x)$: Statistics of importation and exportation in China	S28
Supplementary Table 3 w_i : Weights of each product and those data distributions used for Monte Carlo simulation	S30
Supplementary Table 4 a_i : Average content of resources contained in various products.....	S31
Supplementary Table 6 Resource market prices in recent years (US\$/ton)	S35
Supplementary Table 7 Previous applications of various methods for predicting anthropogenic mineral generation	S36
Supplementary Table 8 Data demand’s comparison of various methods for anthropogenic mineral prediction ⁵⁷	S36
Supplementary Table 9 w_i : Estimation of average PC weight	S37
Supplementary Table 10 w_i : Estimation of average TV weight.....	S37
Supplementary Table 11 Prohibitive imported directory of solid waste and their implemented time	S38
Supplementary Table 12 $I'(x)$: Estimation and range of imported anthropogenic mineral in 2010-2050S43	
Supplementary Table 13 η , β , and L : Parameters of Weibull lifespan distribution and the regulated maximum lifetime.....	S46
Supplementary Table 14 $f(x)$: Lifetime distribution function of all the relevant products	S47
Supplementary References	S50

5) RESULTS: In the section “Dynamic transfer map of valuable resources.” It is unclear why you discuss net addition to in-use stock for only seven of the 21 materials that you were tracking. Presumably this is because of your conclusion that they are the dominant value generators. If this is so, please make this explicit.

Response: All the materials are considered to illustrate the transfer to demand side. This section has been rewritten again to support the title and conclusion.

Meeting potential of anthropogenic mineral supply for resource demand. The future demand of product is theoretically equal to net addition to in-use stock, which is the difference between the consumption and the generated anthropogenic mineral (Eq. 7 and Supplementary Fig. 2). All the relevant materials are also chosen to uncover the future resource demand. Cu, Al, Au, and Pd will maintain the growth until 2030 and afterwards reach the constant (Fig. 4), which is attributed to the flourish of wiring and cable and EEE⁷⁰. Pb demand increased from 0.91Mt in 2000 to 7.75 Mt in 2014⁶⁶, but Pb, Fe, Zn, and rubber are quickly decreasing until 2020, and afterwards almost keep the unchanged range while vehicle and EEE are approaching the final saturation; Co will increase and reach the peak in 2030 because the cathode material in lithium-ion battery is gradually substituted from lithium cobalt oxide to lithium nickel cobalt aluminum oxide and lithium iron phosphate in electric vehicle and consumer electronics⁷¹; Other fifteen materials of future demand have entered a stabilized phase. The plastic, for instance, will always maintain about 6 Mt in 2010-2050 while the falling importation and the increasing production will mingle together.

6) RESULTS: In the presentation of your results, it would be valuable to include a plot that shows the evolution of both the supply (annual domestic anthropogenic mineral generation) and the demand (annual net production) for at least one focal material, hopefully, putting these on the same plot. This would help readers to better visualize the definition of the annual net addition to in-use stock as the gap between these two quantities. I would also suggest calling out the definition of annual net addition in a formula. A formula in this context is needed not because the relationship is complex, but rather because it helps to make the definition of this key result stand out in the manuscript.

Response: Very nice suggestions. We choose eight focal material to illustrate the relationship between the supply and demand. One new session is added to consolidate the article.

Fig. 4 Meeting potential of anthropogenic mineral for demand: (a) Cu; (b) Al; (c) Fe; (d) Plastic; (e) Rubber; (f) Glass; (g) Pb; (h) Zn; (i) Sn; (j) Mg; (k) Co; (l) Au; (m) Ag; (n) Pd; (o) Rh; (p) Pt; (q) In; (r) W; (s) Nd; (t) Dy; (u) La; (v) other RE; (w) Meeting time of various materials.

7) RESULTS: As noted in the title and the abstract, you draw a conclusion that anthropogenic minerals can meet resource demand by 2050. Presumably, this is based on the analysis presented in Figure 4. Because this is such an important conclusion, this paper would be greatly improved if you spent more time discussing these figures and how someone can map these results to the conclusion about meeting resource demand.

Just to explain what I mean here, consider these points. In the way that this section is written up currently, it appears that Annual Net Addition to In-Use Stock is the ultimate metric used to support your conclusion. Presumably as that quantity goes to zero it implies that anthropogenic mineral generation is occurring at a rate comparable to the rate of production. If one looks naively at figure 4a, one might conclude that anthropogenic minerals are only going to be sufficient for Pb because it

is the only one for which the result approaches zero.

Response: Thanks again for your care about the support to title and conclusion. To make it more logic and convincing, we add one new session subtitled Meeting potential of typical anthropogenic mineral material for resource demand (as given above). I think it could be smooth to support the title and conclusion. Additionally, we polish again the deep implications to reinforce the conclusion. Please check them up.

Globally, Au, Ag, and In in remaining underground reserve can only sustain demand for 36 years, 9 years, and 4 years at half of U.S. per capita consumption rates.⁷³ Their exploration and mining will be far harder and more costly owing to the increasing depth of exploration and the descending grade.^{74, 75} Most anthropogenic base metal minerals have the higher grade and purity (as the pure metal or alloy) than the general natural minerals. Eventually, anthropogenic minerals will play an essential role for supply security and for overall sustainability objectives. Although some specific uses of metals require specific properties that are more difficult to obtain from secondary source metals⁷⁶, refinement and metallurgical processes are constantly being improved for resource recovery.⁷⁷ Investing in research and infrastructure to allow for anthropogenic mineral recovery can be an effective approach to relieving this resource shortage bottleneck,^{22, 78, 79} while China relies on imports for many primary minerals⁸⁰. Meanwhile, the increase in the amount of anthropogenic mineral, as shown by our data, suggests more economic viability in harnessing this reserve. At the same time the environmental and social risks associated with harnessing anthropogenic minerals should also be considered alongside those risks for conventional primary mining.

Reviewer #3

1. Comment:

Dear editor and authors,

I have now had time to read and consider this manuscript in detail. In this document, I am providing some general comments, with more specific comments in the PDF of the article and supplement which I upload with this review.

With this contribution, the authors set out on an ambitious mission to not only assess the material flows from various waste products in China in the future (up to 2050), but they also purport to compare the supply potential for different metals and other materials from these waste streams to the total raw materials demand of China. They conclude in the abstract, with their own words that:

“All the obtained results demonstrate that anthropogenic material supply can become the dominant source to meet China’s mineral demand around 2050.” (lines 12 – 14)

While the general subject of this manuscript is certainly interesting, it is not suitable for publication in Nature Communications in its current state. Not only is the quality of the presentation considerably below the standard of the journal, but there are also substantial flaws in the analysis which mean that the conclusion I quoted from the abstract above is certainly not supported by the facts and analysis presented by the authors. I explain in more detail why this is below.

Response: Many thanks for your positive assessment and point-out flaws. We appreciated you for your valuable time input.

We understood well your meanings and suggestions. So we have replenished a great number of new data, added some paragraphs, and polished the logic to improve the article. We believe the new edition can answer the above questions.

The specific comments in the original PDF article have been responded in detail as the new annotation in the same document. We also upload it in the second submission. Please check up.

2. Comment:

Style and quality of presentation

While I understand that English is not the first language of the first author, I found this manuscript extremely difficult to read on account of the quality of its descriptions. Not only does it use excessive

phraseology which renders some sentences virtually unintelligible, but there are also logical gaps and wrongly used terminology that make the reasoning of the authors hard to follow (e.g. lines 41 to 43). As another example of this I would like to use the term “stocks” which is generally taken mean a cumulative quantity. However, the authors use it to mean annual quantities e.g. of waste generated. I only realized this towards the end of the manuscript and got very confused when trying to make sense of figures 2 and 3. I could quote more examples. I am actually surprised that such large deficiencies have not been corrected by the second author, whose English language skills I would have expected to be much better, given that he has held teaching positions at academic institutions in both Australia and the US.

Response: Thanks again for your pointing-out. We have improved the sentences you highlighted. We also polished again all the texts with the language improvement. Meanwhile, a detailed terminologies definition and boundary, including anthropogenic mineral, stock, and in-use stock, is added in the supplementary text 2 to clarify the difference and boundary.

Evolution of valuable resources in anthropogenic mineral. When some products reach their EoL, a large quantity of valuable resources is inevitably encased in anthropogenic mineral⁶⁴, regardless of whether in hibernating stock or not.^{9,65} The amount of encased eleven base materials (e.g., Cu, Al, Fe, Co, Pb, Zn, Sn, Mg, plastics, rubber, and glass), five precious metals (e.g., Au, Pd, Ag, Pt, and Rh), two rare metals (e.g., In and W), and five rare earths (including Nd, Dy, La, Y, and Eu) in yearly-generated anthropogenic mineral can be easily determined (Supplementary Fig. 14). Almost all mineral resources encased in WEEE, ELV, and WWC have been constantly growing since 2010, and can be expected to continue to rise, at least until 2050.

Totally, for base materials, eight metals, except Pb, maintain the increasing tendency despite of a rapid decline of importation. In 2010, the mineral resources of Cu, Al, Fe, Zn, Co, Sn, and Mg were approximately 7.03 Mt, 4.46 Mt, 14.69 Mt, 0.78 kt, 5.82 kt, 0.16 kt, and 46 kt, respectively, but they will rise to 28.52 Mt, 16.35 Mt, 82.63 Mt, 10.65 kt, 23.49 kt, 2.23 kt, and 441 kt on average by 2050, respectively (Fig. 2). The amount of Co and Fe will lead in the increasing rate among all the base materials due to the dramatic and continuing boom of battery and vehicle. The popularity of display substitution and the build-up of vehicle, in particular used with lead-acid battery, have resulted in a peak of 4.3 Mt for Pb generation around 2020, verified by the previous studies^{66,67}. Simultaneously, the amount of plastic, glass, and rubber will increase around 55%-, 8-, and 7.7-fold from 2010 to 2050 (Fig. 2).

The total precious metals are always keeping the growth trend in the year of 2010-2050. The amounts of Au, Pd, Ag, Rh, and Pt were only 89.95, 92.98, 473.84, 2.98, and 6.96 tons, respectively in 2010, but they will grow roughly 15-, 19-, 14-, 8-, and 8-fold in 2050 (Fig. 2c). Actually, around global 85% Rh, 50% Pd, and 43% Pt were used in automobile catalyst scraps.⁶⁸ The same ascending trend in total rare metals and rare earth remains as the other metals. The amounts were approximately 0.15 kt, 26.75t, 0.44 kt, and 0.92 kt for In, W, Nd, and other rare earth, respectively, but they will rise to 2.04 Mt, 366 tons, 5.38 kt, and 12.52 kt by 2050, respectively (Fig. 2).

Supplementary Session 1: Terminologies definition and boundary

(1) Anthropogenic mineral, nature mineral, and solid waste (Supplementary Fig. 1)

Anthropogenic mineral, or called urban mineral, belong to solid waste, is the recycled resources characterizing the product with the orderly physical structure and fixed shape under the condition of human activity.

Nature mineral is the underground solid resource generated by the naturally occurring process.

Solid waste is any garbage or refuse, sludge from a wastewater treatment plant, water supply treatment plant, or air pollution control facility and other discarded material, resulting from industrial, commercial, mining, and agricultural operations, and from community activities.

(2) New scrap and old scrap (Supplementary Fig. 1)

New scrap is the solid waste generated from the manufacturing process.

Old scrap is the solid waste generated from end of life product after consumption or utilization, including the anthropogenic mineral.

(3) Primary resources and secondary resources (Supplementary Fig. 1)

Primary resources called nature resources (mineral), are resources that exist without actions of humankind.

Secondary resources are the resources by anthropogenic activity, consisting of new scrap and old scrap.

Supplementary Fig. 1 Boundary of anthropogenic mineral, natural mineral, primary resources, secondary resources, solid waste, new scrap, and old scrap

(4) Product, resource, and material¹

Product is a good consisting of a bundle of tangible and intangible attributes that satisfies consumers.

Resource stands for the useful matters, consisting of valuable material.

Material is the element, constituent, or substance of which something is composed or can be made.

(5) Urban mining and virgin mining²

Urban mining is the process recycling resources from anthropogenic mineral, consisting of dismantling, crushing, separation, and deep recovery using hydrometallurgy, pyrometallurgy, or biological treatment.

Virgin mining is the process to extract resources from natural mineral, consisting of crushing, separation, and deep recovery using hydrometallurgy, pyrometallurgy, or biological treatment.

(6) Flow, stock, in-use stock, and net addition to in-use stock (Supplementary Fig. 2)^{1, 3}

A flow is the change of a stock over time. The yearly-generated anthropogenic mineral belongs to the flow.

A stock is the reservoir that is accumulated over time by inflows and/or depleted by outflows, which is the totally cumulative quantity.

In-use stock is the cumulative product or resource, which is used or consumed functionally.

Net addition to in-use stock in every year is the difference between the yearly-added consumption and yearly-generated anthropogenic mineral.

3. Another problem is that figures are not well designed and labelled. For instance, Fig. 2a – e and 2f essentially represent the same information in different ways. It would have been better to just use the more compact of these representations. This will also reduce print space. As it is, there is unnecessary redundancy that does not provide any significant gain in information.

Response: We revised all the figures with the estimation error. Overlaps among them have been declined. For instance, Fig. 2f and Fig. 4w only reflect the results on average.

4. Furthermore, the text contains several statements that are simply factually wrong, e.g. that China is largely reliant on imported raw materials (lines 40 and 280/281). Quick reference to the USGS Mineral Commodity Summaries will show that in fact, China is the main producer of many primary metals (Zn, Pb, Ga, Ge, In, Sb, REEs etc.). Another such statement is that recycling is generally cheaper and “better” source for ALL metals than mining (lines 277 / 278). This is similarly untrue. It depends very much on the metal in question. See Reuter et al. (2011) and Allwood (2014). For instance, Al cannot currently be recycled to yield the same material quality as primary Al. Indium is too greatly dispersed in its major applications to warrant recycling. Extraction from primary sources

(zinc ores) is cheaper.

Response: Many thanks for your providing some new insights. All the relevant sentences and address have been revised. We believe the revised edition is better and more scientific. Please check up.

China is not only the world's major manufacturing power, but also one of the largest consumers and exporter of products.¹⁶ Nowadays China has all types of industries in UN International Standard Industrial Classification System and led in 220 products yields of global over 500 industrial products, meeting the demand of the world.¹⁷ China in recent years was the largest importer of minerals bearing solid waste to alleviate domestic resource scarcity. Such features could highlight the unique opportunity for China to measure the potential of anthropogenic mineral supply.^{18, 19}

EEEs and vehicles are the most fashionable aspiration of assets in Chinese households, which are the hallmark of the once-called "Four Big Items" that consumers are aspiring towards (see Supplementary Figs. 3 and 4). Their rapid evolution and popularity since 1970s could reveal a dramatic rise in waste generation and resource consumption. The consumption of a couple of mineral resources has gone through multiple increases²⁰, resulting in a shortage of some important strategic resources and a growth of external dependence.^{21, 22, 23} To meet the future resource demand, resources mining from anthropogenic mineral has become a world concern at the context of circular economy.

...

Globally, Au, Ag, and In in remaining underground reserve can only sustain demand for 36 years, 9 years, and 4 years at half of U.S. per capita consumption rates.⁷³ Their exploration and mining will be far harder and more costly owing to the increasing depth of exploration and the descending grade.^{74, 75} Most anthropogenic base metal minerals have the higher grade and purity (as the pure metal or alloy) than the general natural minerals. Eventually, anthropogenic minerals will play an essential role for supply security and for overall sustainability objectives. Although some specific uses of metals require specific properties that are more difficult to obtain from secondary source metals⁷⁶, refinement and metallurgical processes are constantly being improved for resource recovery.⁷⁷ Investing in research and infrastructure to allow for anthropogenic mineral recovery can be an effective approach to relieving this resource shortage bottleneck,^{22, 78, 79} while China relies on imports for many primary minerals⁸⁰. Meanwhile, the increase in the amount of anthropogenic

mineral, as shown by our data, suggests more economic viability in harnessing this reserve. At the same time the environmental and social risks associated with harnessing anthropogenic minerals should also be considered alongside those risks for conventional primary mining.

5. Methodology and soundness of analysis

While none of the deficiencies listed above would never have led me to reject this manuscript had the authors' analysis been sound, this does, unfortunately, also contain some major deficiencies as indicated above. I will start with the most important problem. The authors make a statement about the potential importance of secondary resources to supply the needs of China in 2050. It is also implicit in the manuscript's title. I would have expected such a strong statement to be supported by a similarly strong analysis of BOTH the expected quantity of waste materials and metals contained within them (as well as estimates of recoverable quantities) AND the expected demand from the Chinese economy. However, the authors only provide a very simplistic analysis of the expected amounts of waste materials, and do not at all consider the demand side anywhere. It is never mentioned. This fact alone invalidates both the title and the supposed major conclusion of this manuscript.

Response: Thanks. We have realized this important problem as indicated by the first reviewer. The title, methods, demand estimation, and meeting potential of anthropogenic have been improved to well support the title and conclusion.

Anthropogenic mineral supply has potential to meet Chinese resource demand

Abstract: An increasingly large quantity of primary mineral resources is being converted into manufactured products and destined in solid waste, which is progressively identified as anthropogenic mineral for potential resource supply. This trend will witness a dramatic shift in China due to a range of policy interventions. China is not only the major producer of consumer products and importer of secondary resource, but also has a major urban footprint. Here we consider three product groups, 30 products, and imports, and map the recycling potential of anthropogenic minerals and 23 types of the capsulated materials by targeting their evolution from 2010 to 2050. Total weight of anthropogenic mineral on average in China reached 39 Mt in 2010, but it will double in 2022 and quadruple in 2045. Stocks of precious metals and rare earths will increase faster than most base materials. The total economic potential in yearly-generated anthropogenic minerals is anticipated to grow markedly from 100 billion US\$ in 2020 to 400 billion US\$ in 2050. Anthropogenic sources of around 20 minerals will be able to meet projected demand by 2050, due to high availability of recycled content and gradual saturation of consumption. Durability

of material usage and the concomitant stock of the anthropogenic resource remain major challenges in determining the viability of this supply in the second half of the coming century.

...

$$D(m) = \sum_{i=1}^{31} \int_{x_0}^m f_i(x) \cdot C_i(x) \cdot w_i dx$$

$$\begin{aligned} &= \sum_{i=1}^{15} [f_i(m-1990) \times C_i(1990) + f_i(m-1991) \times C_i(1991) + K + f_i(1) \times C_i(m-1)] \\ &+ \sum_{i=1}^{15} [f_i(m-1991) \times C_i(1991) + f_i(m-1992) \times C_i(1992) + K + f_i(1) \times C_i(m-1)] \quad (5) \\ &+ [f_i(m-1996) \times C_i(1996) + f_i(m-1997) \times C_i(1997) + K + f_i(1) \times C_i(m-1)] \end{aligned}$$

$$C(x) = P(x) - E(x) + I(x) \quad (6)$$

$$D'(x) = C(x) - D(x) \quad (7)$$

$$W_{m,j} = \sum_{i=1}^{31} \left[\int_{x_0}^m f_i(x) \cdot C_i(x) \cdot w_i dx \cdot c_{ij} \right] \quad (8)$$

$$S_{m,j} = \sum_{i=1}^{31} \{ [C_i(m) - D_i(m)] \cdot w_i \cdot c_{ij} \} \quad (9)$$

$$M = W / S \times 100 \quad (10)$$

where m is the assigned year (2010-2050) for the concerned generation; i is the i^{th} category of product; 31 is total estimated categories for EEE, vehicle, and wiring & cable; x_0 is the initial year of production ($x_0=1990$ for EEEs, $x_0=1991$ for vehicle, and $x_0=1996$ for wiring & cable); $C(x)$ is the consumption quantity of product flow from production, exportation, and importation (in million or ton); $D'(x)$ is the theoretical demand of product (in million or ton); w_i is the weight of i^{th} category of product (in kg or ton); $P(x)$ is the domestic production quantity (in million); $E(x)$ is the exportation of product (in million); $I(x)$ is the importation of product (in million); j is the j^{th} resource category; c_{ij} is the content of the j^{th} resource in the i^{th} category of products; $W_{m,j}$ is the total weight of j^{th} resource in the yearly-generated anthropogenic mineral at the year of m (in ton), which indicates the supply potential of anthropogenic mineral; $S_{m,j}$ is the total weight of j^{th} resource in the yearly-added consumption at the year of m (in ton), which indicates the theoretical demand of resource; and M is defined as the resource supply meeting ratio of anthropogenic mineral for the demand (%).

.....

Meeting potential of anthropogenic mineral supply for resource demand. The future demand of product is theoretically equal to net addition to in-use stock, which is the difference between the consumption and the

generated anthropogenic mineral (Eq. 7). All the relevant materials are also chosen to uncover the future resource demand. Cu, Al, Au, and Pd will maintain the growth until 2030 and afterwards reach the constant (Fig. 4), which is attributed to the flourish of wiring and cable and EEE^1 . Pb demand increased from 0.91Mt in 2000 to 7.75 Mt in 2014², but Pb, Fe, Zn, and rubber are quickly decreasing until 2020, and afterwards almost keep the unchanged range while vehicle and EEE are approaching the final saturation; Co will increase and reach the peak in 2030 because the cathode material in lithium-ion battery is gradually substituted from lithium cobalt oxide to lithium nickel cobalt aluminum oxide and lithium iron phosphate in electric vehicle and consumer electronics³; Other fifteen materials of future demand have entered a stabilized phase. The plastic, for instance, will always maintain about 6 Mt in 2010-2050 while the falling importation and the increasing production will mingle together.

Fig. 4 Meeting potential of anthropogenic mineral for demand: (a) Cu; (b) Al; (c) Fe; (d) Plastic; (e) Rubber; (f) Glass; (g) Pb; (h) Zn; (i) Sn; (j) Mg; (k) Co; (l) Au; (m) Ag; (n) Pd; (o) Rh; (p) Pt; (q) In; (r) W; (s) Nd; (t) Dy; (u) La; (v) other RE; (w) Meeting time of various materials.

We further uncover the supply potential of anthropogenic minerals. With the dramatic rise of anthropogenic reserve and the gradual saturation of material demand, the fully supply from anthropogenic mineral for the resource demand is becoming possible (Fig. 4). Eighteen materials of anthropogenic reserve could meet their demand before 2020, and in 2050 they hopefully provide over two-fold demand. The meeting time of Sn and Pd will be approximately 2030 and 2041, respectively. Although Cu, Al, and Co of anthropogenic reserve cannot meet their demand by 2050, the disparity gap between their demand and anthropogenic minerals will be greatly reduced in the following decades. Consequently, anthropogenic mineral supply has a growing potential to meet the future

resource demand.

6. Nevertheless, a proper quantitative analysis of waste streams and their probable evolution could still have been valuable in its own regard. However, this is not what the authors purportedly set out to represent, nor is their current analysis without flaws. My major points of criticism in this regard center on: 1) extremely simplistic forecasting of waste flows, and 2) insufficient analysis of uncertainties.

Forecasting. To assess the amounts of materials ending up as waste, the authors use a combination of production statistics for the products they investigated (from 2003 – 2013) and their extrapolation into the future in combination with a probabilistic description of likely product lifetimes. My problem here mainly concerns the extrapolation into the future. Based on mostly linear but also sometimes exponential fits (not clear, why two methods are used, and what the justification is in specific cases) to the 11 years or so for which they have data, they simply extrapolate 35 years into the future (!). This is extremely naïve. Not only is not clear that linear growth is a good representation of the likely future development for different product categories, particularly those showing a decrease in production volume over the time-span considered (they would for several products end up negative). But it is also questionable that these trends are robust over a time span that is more than THREE TIMES the range spanned by the input data. Which brings me to the next point: uncertainty analysis.

Response: Thanks for your pointing-out again. We have carefully revised these questions from the methods to results and logic.

7. Uncertainty analysis. While the authors supposedly include a very crude uncertainty analysis, they show corresponding confidence intervals in none of their figures. I would have expected to see these, using the standard 95% reporting method. Furthermore, the authors only consider one major source of uncertainty – the composition of waste materials. However, they ignore an even more substantial one: the very large uncertainties that will result from their forecasting methods. All of their fits to production data will have uncertainties. The further these trends are extrapolated into the future, the

larger these uncertainties become. Given the large degree of extrapolation, it is likely that by 2050 these uncertainties will be so large that definitive statements on available amounts to within a factor 2 or less will be nearly impossible. This needs to be considered.

Response: Good questions. Yes, uncertainty is caused by not only from the data, but also from the method. Regarding the prediction method, we add one paragraph to address how and why for the data regression. In the revised article, uncertainty for data is realized with Monte Carol simulation, and uncertainty for method is given with the error or range of the prediction data. Please check up the revised methods and all the relevant figures.

***Data regression.** We chose a stock-based model^{50, 51} to define the net production of EEE, vehicle, wiring and cable. The net production of any given products from Eq. 1 was assumed to be both consumed and discarded in China, and its future consumption was then determined using a time-step method based on data regression. Ideally, the Logistic equation is the feasible to enable the forecasting future consumption. It is characterized of the rapid growth at the start stage, stable growth at the middle stage, and slow growth for a constant at the late stage, which is approximately expressed by exponential function, linear function, and constant value, respectively. Additionally, the decline of production and consumption commonly occurs while the product is replaced.*

.....

***Uncertainty analysis.** Both the method and the input data will affect the uncertainty of estimated results. The method, in particular from data regression, will be identified with the feasible error. The data including weight and average content will be assessed with uncertainty analysis. Here, a Monte Carlo simulation (10^5 iterations) was conducted to obtain final estimates of flows and their uncertainties in this study. Based on the relevant collected data, and those distributions illustrated in Supplementary Tables 3, 4, and 14, uncertainties of total anthropogenic mineral weight and yearly-added resource stock (e.g., Cu and Al) in 2017 & 2030, for instance, are performed and presented in Supplementary Fig. 8b. Similarly, the resources stock from Monte Carlo simulation can also cover the forecasting results at the maximum probability interval.*

.....

***Anthropogenic mineral generation.** We firstly pre-mined the available data for further estimation of*

domestic generation of anthropogenic mineral: (1) Lifetime distribution function of all the relevant products is determined by Supplementary Text 3 and Table 15; (2) Data regression is enabled with Supplementary Figs. 10-12 for the annual net production and imported scraps; and (3) the incremental error or range are configured for the estimation of future consumption and importation (Supplementary excel table and Table 13). Accordingly, domestic generation of WEEE, ELV, and WWC are uncovered (Fig. 1a and Supplementary Fig. 13)...

8. Results and conclusions

Given the many reservations I have already voiced above, I do not think it is sensible to comment in detail on these sections. They will at any rate require a complete rewriting to account for the changes made elsewhere in the manuscript to address my recommendations (below).

Response: We rewrote the results and conclusions to describe all the obtained results. Please check up the new edition.

9. Suggestions for improvement

Following on from my comments above, I have two major suggestions for improvement. First, I suggest the authors rethink the purpose of this publication. With their current analysis, neither title nor major conclusion are justified. If they wish to make these conclusions, they also need to include an assessment of the demand side for China up to 2050. It may be easier to just focus on supply.

Second, I suggest the authors incorporate the fitting uncertainty of their linear (and other) regression analysis into their uncertainty assessment. I encourage them to try out different functional forms. I expect they will be surprised at how large these uncertainties become decades into the future.

Third, I suggest the authors give the revised text that incorporates the changes suggested above a complete overhaul to improve the quality and style of their language, and ensure a concise and coherent presentation, ideally with the help of a native speaker or a professional language editing service. This should ensure that the manuscript meets the quality requirements of the journal and is easier on readers.

Response: We appreciate you with very valuable comments and suggestions. They indeed raise the research article. The logic and contribution have been examined again from introduction to results

and conclusion. The demand side is highlighted to compare with the generated anthropogenic mineral. Thus, the results about anthropogenic mineral meeting potential are given and polished to support the new title and conclusion. Please check them up. We believe the new edition is better. Again, the fitting uncertainty of their linear (and other) regression has been deliberately improved with million data recalculating of error and range.

10. References cited

Allwood JM (2014) Squaring the circular economy: The role of recycling within a hierarchy of material management strategies. In: Worrell E, Reuter M (Eds) Handbook of Recycling, Elsevier, Amsterdam-New York-London.

Reuter MA (2011) Limits of design for recycling and “sustainability”: A review. Waste Biomass Valor 2:183-208.

Response: We read these two references and put the idea into the revised article.

Other information and data have been updated and improved, please find the full manuscript. MANY THANKS for your valuable comments and suggestions!

Reviewers' comments:

Reviewer #1 (Remarks to the Author):

****General Comments**

This revision has made important improvements in language, clarity of presentation, and rationalization of the scope and conclusions. Now that clarity is improved it is easier to isolate a few topic whose ambiguity would likely generate confusion to the reader. As such, I believe further revision is warranted to ensure that readers clearly understand your findings. In particular, you need to

- 1) Provide a clear definition of anthropogenic minerals and stick with that term throughout
- 2) Make it clear that the scope of demand that you are attempting to satisfy comprises the same products that are found within the anthropogenic mineral
- 3) Check your use of the word demand through the manuscript
- 4) Check your definition of the meeting ratio

I encourage you to put in the effort to address these issues.

****Terminology around anthropogenic materials and its associated products**

*****Definition of anthropogenic minerals**

In your rebuttal you indicate that the term "anthropogenic mineral" has a fixed definition in China and you imply that this scope is what you mean when you use that term in the manuscript. (from the rebuttal) "In China, the government body, National Development and Reform Commission in 2010 proposed a distinct concept of anthropogenic mineral defined as the recycled iron, non-ferrous metals, precious metals, plastic, or rubber material lain in three large groups of waste products such as electrical and electronic equipment (EEE), vehicles, and wires and cables,"

Some changes will need to be made to establish that formality in the manuscript itself. Consider this example: On the first page, you use the term "anthropogenic sources" but then follow it with a definition that is reminiscent of the one you provide in the rebuttal document.

If you are going to define "anthropogenic mineral" as a formal concept, then you need to use that term consistently.

To begin this process, I would recommend that these sentences

"29 ...The scope of such anthropogenic sources is strictly regulated by the Chinese government and comprises recycled iron, non-ferrous metals, precious metals, plastic, or rubber material embedded in three groups of waste products such as electrical and electronic equipment (EEE, all the abbreviations and acronyms are provided at Supplementary Information Section 2), vehicles, and wires and cables,"

be changed to "One such waste stream is strictly regulated by the Chinese government. It is referred to as "anthropogenic mineral" (AM) and was defined by the National Development and Reform Commission in 2010 to comprise the iron, non-ferrous metals, precious metals, plastic, or rubber material found within three waste products: electrical and electronic equipment (EEE), vehicles, and wires and cables."

*****Scope of the products whose consumption needs are met by AM**

This ambiguity of terminology follows you into the discussion of your results. In particular, I believe that readers will be confused about the scope of what consumption you are comparing to your AM supply.

Consider for example the text in lines 308-312

"We further uncover the supply potential of anthropogenic minerals. With the dramatic rise of anthropogenic reserve and the gradual saturation of material demand, the fully supply from anthropogenic mineral for the resource demand is becoming possible (Fig. 4). Eighteen materials of anthropogenic reserve could meet their demand before 2020, and in 2050 they hopefully provide over two-fold demand."

I believe that a reader could easily interpret these sentences to mean that AM could satisfy consumption for all sectors for selected materials within AM. I am fairly confident that is not what you mean here. Instead I believe that you mean that resources with AM could satisfy consumption of selected materials in EEE, vehicles, and wire and cable – only. This is an interesting conclusion, but its scope should be made clear.

Consider this also for the title of the article "Anthropogenic mineral supply has potential to meet Chinese resource demand". I believe that you mean it has the potential to meet demand for electronics, vehicles, and wire and cable. I would be valuable to come up with a concise way that would make that more clear.

**Consumption, Demand and the Meeting Ratio

I am concerned about the consistency of the use of the term demand and more broadly about its specific definition in the manuscript. Consider this example from Lines 283-285

"The future demand of product is theoretically equal to net addition to in-use stock, which is the difference between the consumption and the generated anthropogenic mineral "

This sentence well captures a challenge in communicating your work. Through examination of supplementary figure 2, it appears that you are defining demand to be the difference between consumption and AM generation. Essentially, it is the quantity of material that must be supplied by something other than waste to satisfy consumption. Assuming that I am understanding that correctly, in my opinion, the use of demand in this way is confusing. I realize that notionally you can define a term and use it in your paper. However, when words are used differently outside of your paper, this practice can easily lead to confusion.

Colloquially and even within much of the academic community (the notable exceptions being economics and electricity power networks), the two terms demand and consumption are often treated as synonyms. Within the economics community, demand is something that cannot be directly observed but reflects an individual or groups preference for some good (generically referred to as some utility) at a set of prices. In that same field, consumption is that action that is actually observed.

I believe that your use of consumption here is sound. I would recommend however selecting another word than demand. Maybe give it a clearly technical name "waste resource shortfall" (WRS) so that readers no that they need to examine how this is being defined in this case. Maybe the simplest option would be to continue to refer to it as the net addition to stocks and not relabel it as demand.

This leads me to back up and ask about why define the meeting ratio or at least why define it as it is. It would seem simpler to track either S (equation 9) and follow when it goes to zero or to define the meeting ratio as D or W divided by C .

I simply don't follow the logic of the meeting ratio being $W / (C-D)$. You define W to be "weight of jth resource in the yearly-generated anthropogenic mineral at the year of m (in ton), which indicates the supply potential of anthropogenic mineral;" This seems quite reasonable. But you define S to be "total weight of jth resource in the yearly-added consumption at the year of m " Comparing these two

doesn't make sense to me. Why compare the total generated waste to the "added" consumption. Should we compare total generation to total consumption?

This brings me back to the results plotted in Figure 4. Is this resource demand the same as the demand you defined on lines 283-285? There is (and in supplementary figure 2) it is defined as equal to net additions to stock. If net additions to stock stay flat and positive (e.g., figure 4c), wouldn't this by definition mean that you are not able to meet all of consumption with the AM? Please clarify this. More to the point, any quantity that is plotted in figure 4 should be able to be matched with one of the equations 5 to 10 and should use the same label.

**Specific comments

Lines 59 – 61

"Since waste electrical and electronic equipment (WEEE or e-waste), end-of-life vehicle (ELV), and waste wiring and cable (WWC) can mainly constitute an anthropogenic mineral"

What does it mean "can mainly constitute"? Are there other things in AM?

Lines 115-116

"China's anthropogenic mineral can be sourced both to domestically consumed products and imported waste"

I think that you mean "China's AM can derive from both domestically consumed products and imported waste"

Lines 226 – 229

"Totally, the weight of the yearly-generated anthropogenic mineral generation in China was estimated for 40 Mt in 2010. Driven by the large expansion of WEEE, ELV, and WWC, total generation weight will reach 71 Mt in 2020, 101 Mt in 2030, and 176 Mt in 2050 (Fig. 1d). The average annual amount in 2010-2050 will be 3.4 Mt, and over one half will be provided by ELV"

The first half of this paragraph is confusing. You note yearly-generated AM is 40Mt in 2010 and this figure grows dramatically to 176Mt by 2050. What then is the "average annual amount in 2010-2050". It can't be the average yearly-generated AM, because even the lowest year (2010) is well above this average.

Randolph Kirchain
kirchain@Mit.edu

Reviewer #2 (Remarks to the Author):

Having read the revised version of the manuscript, I have the following further comments, compared to my last review:

- 1) Language quality has been improved but is still not up to the standard of the journal.
- 2) The authors now include estimates of future demand in Fig. 4 and compare them with anthropogenic reserves. Purportedly, this supports their conclusion that anthropogenic materials will be sufficient to cover demand. However, this is not what the comparison shows. Reserves are a cumulative quantity, while demand is a rate. What needs to be compared here are the rate of waste generation and demand. I expect they will not be equal for a long time, particularly under scenarios of

increasing demand (which would necessarily accompany increased consumption and increased future waste generation). Furthermore - not all of the material contained in the waste will be recoverable. This is, however, another implicit assumption the authors make without much further discussion. It is unrealistic, particularly for elements like In, Ge, Pt etc. that are highly dispersed in industrial products, and may even be lost in use (Pt).

I would therefore contend that the major conclusion of the paper is still not supported by the authors' analysis, and does in fact remain untenable. The quality of the presentation is also still not up to standard and difficult to follow. Fig. 4, for instance is extremely difficult to read, since there is no indication in the panels themselves, of which element they specifically refer to.

Hence, I must uphold my recommendation from last time, that this manuscript still requires major revisions.

Reviewer #1

1. **General Comments

This revision has made important improvements in language, clarity of presentation, and rationalization of the scope and conclusions. Now that clarity is improved it is easier to isolate a few topic whose ambiguity would likely generate confusion to the reader. As such, I believe further revision is warranted to ensure that readers clearly understand your findings. In particular, you need to

- 1) Provide a clear definition of anthropogenic minerals and stick with that term throughout
- 2) Make it clear that the scope of demand that you are attempting to satisfy comprises the same products that are found within the anthropogenic mineral
- 3) Check your use of the word demand through the manuscript
- 4) Check your definition of the meeting ratio

I encourage you to put in the effort to address these issues.

Response: Many thanks again for your positive comments and valuable suggestions. All the suggestions have been fully considered to improve the article.

- 1) In the newly revised article, a clear definition of anthropogenic minerals has been given and the term is used consistently through the full article. please check it up.
- 2) We improve the address about the scope of anthropogenic mineral.
- 3) The word demand has been checked through the article.
- 4) The definition of meeting ratio has been checked.

2. **Terminology around anthropogenic materials and its associated products

***Definition of anthropogenic minerals

In your rebuttal you indicate that the term “anthropogenic mineral” has a fixed definition in China and you imply that this scope is what you mean when you use that term in the manuscript. (from the rebuttal) “In China, the government body, National Development and Reform Commission in 2010 proposed a distinct concept of anthropogenic mineral defined as the recycled iron, non-ferrous metals, precious metals, plastic, or rubber material lain in three large groups of waste products such as electrical and electronic equipment (EEE), vehicles, and wires and cables,”

Some changes will need to be made to establish that formality in the manuscript itself. Consider this

example: On the first page, you use the term “anthropogenic sources” but then follow it with a definition that is reminiscent of the one you provide in the rebuttal document.

If you are going to define “anthropogenic mineral” as a formal concept, then you need to use that term consistently.

To begin this process, I would recommend that these sentences “ 29 ...The scope of such anthropogenic sources is strictly regulated by the Chinese government and comprises recycled iron, non-ferrous metals, precious metals, plastic, or rubber material embedded in three groups of waste products such as electrical and electronic equipment (EEE, all the abbreviations and acronyms are provided at Supplementary Information Section 2), vehicles, and wires and cables,”

be changed to “One such waste stream is strictly regulated by the Chinese government. It is referred to as “anthropogenic mineral” (AM) and was defined by the National Development and Reform Commission in 2010 to comprise the iron, non-ferrous metals, precious metals, plastic, or rubber material found within three waste products: electrical and electronic equipment (EEE), vehicles, and wires and cables.”

Response: Many thanks for nice recommendation. We agreed. These sentences have been improved to clarify the terminology and definition of anthropogenic mineral.

... However, the shortening useful life expectancy of the product, driven by rapid innovation, miniaturization and affordability, and an increasingly anthropogenic metabolism have led to a major increase in the accumulation of waste product, which could potentially be classified as “anthropogenic mineral” (AM, some relevant terminologies definition and boundaries are provided in Supplementary Information Section 1).^{3, 5} One such waste stream is strictly regulated by the Chinese government. It is referred to as AM and was defined by the National Development and Reform Commission in 2010 to comprise the iron, non-ferrous metals, precious metals, plastic, or rubber material found within three waste products: electrical and electronic equipment (EEE, all the abbreviations and acronyms are also provided at Supplementary Information Section 2), vehicle, and wire & cable.^{6, 7} These are identified as the core scope of AM not only in China, but also in many industrial nations.^{8, 9, 10, 11}

Critical raw materials have also been sinking into AM reserves, while accelerating the depletion of virginal minerals.^{12, 13, 14, 15} On the other hand, China is not only the world’s major manufacturing power, but also one of the largest consumers and exporter of products.¹⁶ Nowadays China has all types

of industries in UN International Standard Industrial Classification System and led in 220 products yields of global over 500 industrial products.¹⁷ China in recent years was also the largest importer of secondary resource to alleviate domestic material scarcity. Such features could highlight the unique opportunity for China to measure the potential of AM supply.^{18, 19}

3. ***Scope of the products whose consumption needs are met by AM

This ambiguity of terminology follows you into the discussion of your results. In particular, I believe that readers will be confused about the scope of what consumption you are comparing to your AM supply.

Consider for example the text in lines 308-312

“ We further uncover the supply potential of anthropogenic minerals. With the dramatic rise of anthropogenic reserve and the gradual saturation of material demand, the fully supply from anthropogenic mineral for the resource demand is becoming possible (Fig. 4). Eighteen materials of anthropogenic reserve could meet their demand before 2020, and in 2050 they hopefully provide over two-fold demand.”

I believe that a reader could easily interpret these sentences to mean that AM could satisfy consumption for all sectors for selected materials within AM. I am fairly confident that is not what you mean here. Instead I believe that you mean that resources with AM could satisfy consumption of selected materials in EEE, vehicles, and wire and cable – only. This is an interesting conclusion, but its scope should be made clear.

Consider this also for the title of the article “Anthropogenic mineral supply has potential to meet Chinese resource demand”. I believe that you mean it has the potential to meet demand for electronics, vehicles, and wire and cable. I would be valuable to come up with a concise way that would make that more clear.

Response: Many thanks again. Your understanding is right. We clarify some sentences from title, abstract and the equation to a couple of texts. Please check them up.

Anthropogenic mineral supply through a circular economy approach has potential to meet Chinese resource consumption

Abstract: An increasingly large quantity of primary mineral resource is being converted into

manufactured products and destined for solid waste disposal. This material can be reclassified as “anthropogenic mineral reserves” and be a potential source of metals for a range of manufacturing uses. China is implementing a range of policy interventions which can lead to such a classification that will raise the profile of recycling programs as a means of metal supply. China is not only a major producer of consumer products and importer of secondary metals, but also has a major urban infrastructure footprint. Here we consider three product groups, 30 products, and imports, and map the recycling potential of anthropogenic minerals and 23 types of the capsulated materials by targeting their evolution from 2010 to 2050. Total weight of anthropogenic minerals on average in China reached 39 Mt in 2010, but it will double in 2022 and quadruple in 2045. Stocks of precious metals and rare earths will increase faster than most base materials. The total economic potential in yearly-generated anthropogenic minerals is anticipated to grow markedly from 100 billion US\$ in 2020 to 400 billion US\$ in 2050. Anthropogenic minerals of around 20 materials will be able to meet projected consumption of three product groups by 2050, due to high availability of recycled content and gradual saturation of consumption. Durability of material usage and the concomitant stock of the anthropogenic minerals remain major challenges in determining the viability of this supply in the second half of the coming century.

...

$$M = \frac{D}{C} \times 100 = \frac{D}{D' - D} \times 100 \quad (10)$$

... and M is defined as the meeting ratio of AM supply for the resource demand from the three product groups (%).

...

Meeting potential of AM supply for future resource consumption of three product groups. The future consumption of product is theoretically equal to net addition to in-use stock, which is the difference between the demand and the generated AM (Eq. 7 and Supplementary Fig. 2). All the relevant materials are also chosen to uncover the future resource consumption imposed by EEE, vehicle, and wire & cable. Cu, Al, Au, and Pd will maintain the growth until 2030 and afterwards reach the constant (Fig. 4), which is attributed to the flourish of wiring and cable and EEE⁷⁰. Pb demand increased from 0.91Mt in 2000 to 7.75 Mt in 2014⁶⁶, but Pb, Fe, Zn, and rubber are quickly decreasing until 2020, and afterwards almost keep the unchanged range while vehicle and EEE are approaching the final

saturation; Co will increase and reach the peak in 2030 because the cathode material in lithium-ion battery is gradually substituted from lithium cobalt oxide to lithium nickel cobalt aluminum oxide and lithium iron phosphate in electric vehicle and consumer electronics⁷¹; Other fifteen materials of future consumption have entered a stabilized phase. The plastic, for instance, will always maintain about 6 Mt in 2010-2050 while the falling importation and the increasing production will mingle together.

...

We further uncover the supply potential of AMs. With the dramatic rise of AM generation and the gradual saturation of material consumption, the potential supply from AM is becoming possible to overtake the resource consumption of three product groups (Fig. 4). Although we are currently still far from a closed-loop society owing to low recycling rate⁷², a rapid advancement is indeed arising for regulation, policy, and technology of circular economy and urban mining. The highly-efficient collection and the cutting-edge recycling will significantly enhance the recycling rate in the future. Thus, if substantial recycling, eighteen materials of AM could meet their demand before 2020, and in 2050 they probably provide over two-fold consumption. The meeting time of Sn and Pd will be approximately 2030 and 2041, respectively. Although Cu, Al, and Co of AM cannot meet their potential consumption by 2050, the disparity gap between their consumption and AMs will be greatly reduced in the following decades. Consequently, AM supply has a growing potential to meet their future resource consumption.

4. **Consumption, Demand and the Meeting Ratio

I am concerned about the consistency of the use of the term demand and more broadly about its specific definition in the manuscript. Consider this example from Lines 283-285

“The future demand of product is theoretically equal to net addition to in-use stock, which is the difference between the consumption and the generated anthropogenic mineral ”

This sentence well captures a challenge in communicating your work. Through examination of supplementary figure 2, it appears that you are defining demand to be the difference between consumption and AM generation. Essentially, it is the quantity of material that must be supplied by something other than waste to satisfy consumption. Assuming that I am understanding that correctly, in my opinion, the use of demand in this way is confusing. I realize that notionally you can define a

term and use it in your paper. However, when words are used differently outside of your paper, this practice can easily lead to confusion.

Colloquially and even within much of the academic community (the notable exceptions being economics and electricity power networks), the two terms demand and consumption are often treated as synonyms. Within the economics community, demand is something that cannot be directly observed but reflects an individual or groups preference for some good (generically referred to as some utility) at a set of prices. In that same field, consumption is that action that is actually observed.

I believe that your use of consumption here is sound. I would recommend however selecting another word than demand. Maybe give it a clearly technical name “waste resource shortfall” (WRS) so that readers no that they need to examine how this is being defined in this case. Maybe the simplest option would be to continue to refer to it as the net addition to stocks and not relabel it as demand.

This leads me to back up and ask about why define the meeting ratio or at least why define it as it is. It would seem simpler to track either S (equation 9) and follow when it goes to zero or to define the meeting ratio as D or W divided by C.

I simply don't follow the logic of the meeting ratio being $W / (C-D)$. You define W to be “weight of jth resource in the yearly-generated anthropogenic mineral at the year of m (in ton), which indicates the supply potential of anthropogenic mineral;” This seems quite reasonable. But you define S to be “total weight of jth resource in the yearly-added consumption at the year of m” Comparing these two doesn't make sense to me. Why compare the total generated waste to the “added” consumption. Should we compare total generation to total consumption?

This brings me back to the results plotted in Figure 4. Is this resource demand the same as the demand you defined on lines 283-285? There is (and in supplementary figure 2) it is defined as equal to net additions to stock. If net additions to stock stay flat and positive (e.g., figure 4c), wouldn't this by definition mean that you are not able to meet all of consumption with the AM? Please clarify this. More to the point, any quantity that is plotted in figure 4 should be able to be matched with one of the equations 5 to 10 and should use the same label.

Response: Much appreciated for your careful check, valuable sharing, and positive suggestions. The discussion about Consumption, Demand and the Meeting Ratio has been improved thoroughly. Main revised viewpoints are identified here:

- The use of consumption is indeed better than the demand so that ‘demand’ is replaced by ‘consumption’ in the revised article.
- Demand amount (D') – AM amount (D) = Actual consumption/using amount (C)
- Meeting ratio (M) = $\frac{\text{Waste amount } (D)}{\text{Consumption amount } (C)} \times 100\% = \frac{D}{D'-D} \times 100\%$
- Yearly-generated total AM amount should be compared to yearly consumption amount, not yearly-added consumption amount.

Therefore, we revised all the relevant texts from the title, equation, tables, and figures, to sentences.

Anthropogenic mineral supply through a circular economy approach has potential to meet Chinese resource consumption

Abstract: An increasingly large quantity of primary mineral resource is being converted into manufactured products and destined for solid waste disposal. This material can be reclassified as “anthropogenic mineral reserves” and be a potential source of metals for a range of manufacturing uses. China is implementing a range of policy interventions which can lead to such a classification that will raise the profile of recycling programs as a means of metal supply. China is not only a major producer of consumer products and importer of secondary metals, but also has a major urban infrastructure footprint. Here we consider three product groups, 30 products, and imports, and map the recycling potential of anthropogenic minerals and 23 types of the capsulated materials by targeting their evolution from 2010 to 2050. Total weight of anthropogenic minerals on average in China reached 39 Mt in 2010, but it will double in 2022 and quadruple in 2045. Stocks of precious metals and rare earths will increase faster than most base materials. The total economic potential in yearly-generated anthropogenic minerals is anticipated to grow markedly from 100 billion US\$ in 2020 to 400 billion US\$ in 2050. Anthropogenic minerals of around 20 materials will be able to meet projected consumption of three product groups by 2050, due to high availability of recycled content and gradual saturation of consumption. Durability of material usage and the concomitant stock of the anthropogenic minerals remain major challenges in determining the viability of this supply in the second half of the coming century.

...

Data regression. We chose a stock-based model^{50, 51} to define the net production of EEE, vehicle, wiring

and cable. The net production of any given products from Eq. 1 was assumed to be both consumed and discarded in China, and its future demand was then determined using a time-step method based on data regression. Ideally, the Logistic equation is the feasible to enable the forecasting future demand...

$$D(m) = \sum_{i=1}^{31} \int_{x_0}^m f_i(x) \cdot D'_i(x) \cdot w_i dx = D(m)_{\text{WEEE}} + D(m)_{\text{ELV}} + D(m)_{\text{WWC}}$$

$$= \sum_{i=1}^{15} [f_i(m-1990) \times D'_i(1990) + f_i(m-1991) \times D'_i(1991) + \dots + f_i(1) \times D'_i(m-1)] \\ + \sum_{i=1}^{15} [f_i(m-1991) \times D'_i(1991) + f_i(m-1992) \times D'_i(1992) + \dots + f_i(1) \times D'_i(m-1)] \quad (5)$$

$$+ [f_i(m-1996) \times D'_i(1996) + f_i(m-1997) \times D'_i(1997) + \dots + f_i(1) \times D'_i(m-1)]$$

$$D'(x) = P(x) - E(x) + I(x) \quad (6)$$

$$C(x) = D'(x) - D(x) \quad (7)$$

$$D_{m,j} = \sum_{i=1}^{31} \left[\int_{x_0}^m f_i(x) \cdot D'_i(x) \cdot w_i dx \cdot c_{ij} \right] \quad (8)$$

$$C_{m,j} = D'_{m,j} - D_{m,j} = \sum_{i=1}^{31} \{ [D'_i(m) - D_i(m)] \cdot w_i \cdot c_{ij} \} \quad (9)$$

$$M = \frac{D}{C} \times 100 = \frac{D}{D' - D} \times 100 \quad (10)$$

where m is the assigned year (2010-2050) for the concerned generation; i is the i^{th} category of product; 31 is total estimated categories for EEE, vehicle, and wiring & cable; x_0 is the initial year of production ($x_0=1990$ for EEEs, $x_0=1991$ for vehicle, and $x_0=1996$ for wiring & cable); $D'(x)$ is the demand quantity of product flow from production, exportation, and importation (in million or ton); $C(x)$ is the theoretical consumption of product (in million or ton); w_i is the weight of i^{th} category of each product (in kg or ton); $P(x)$ is the domestic production quantity (in million); $E(x)$ is the exportation of product (in million); $I(x)$ is the importation of product (in million); j is the j^{th} resource category; c_{ij} is the content of the j^{th} resource in the i^{th} category of products; $W_{m,j}$ is the total weight of j^{th} resource in the yearly-generated AM at the year of m (in ton), which indicates the supply potential of AM; $C_{m,j}$ is the total weight of j^{th} resource in the consumption at the year of m (in ton), and M is defined as the meeting ratio of a certain resource supply from AM to the resource consumption from the three product groups (%).

5. **Specific comments

Lines 59 – 61

“Since waste electrical and electronic equipment (WEEE or e-waste), end-of-life vehicle (ELV), and waste wiring and cable (WWC) can mainly constitute an anthropogenic mineral”

What does it mean “can mainly constitute”? Are there other things in AM?

Response: The scope of AM differs in many countries and regions. But WEEE, ELV, and WWC are main and common types of AM.

... “anthropogenic mineral” (AM, some relevant terminologies definition and boundaries are provided in Supplementary Information Section 1).^{3, 5} One such waste stream is strictly regulated by the Chinese government. It is referred to as AM and was defined by the National Development and Reform Commission in 2010 to comprise the iron, non-ferrous metals, precious metals, plastic, or rubber material found within three waste products: electrical and electronic equipment (EEE, all the abbreviations and acronyms are also provided at Supplementary Information Section 2), vehicle, and wire & cable.^{6, 7} These are identified as the core scope of AM not only in China, but also in many industrial nations.^{8, 9, 10, 11}

6. Lines 115-116

“China’s anthropogenic mineral can be sourced both to domestically consumed products and imported waste”

I think that you mean “China’s AM can derive from both domestically consumed products and imported waste”

Response: Right, revised.

7. Lines 226 – 229

“Totally, the weight of the yearly-generated anthropogenic mineral generation in China was estimated for 40 Mt in 2010. Driven by the large expansion of WEEE, ELV, and WWC, total generation weight will reach 71 Mt in 2020, 101 Mt in 2030, and 176 Mt in 2050 (Fig. 1d). The average annual amount in 2010-2050 will be 3.4 Mt, and over one half will be provided by ELV”

The first half of this paragraph is confusing. You note yearly-generated AM is 40Mt in 2010 and this figure grows dramatically to 176Mt by 2050. What then is the “average annual amount in 2010-2050”. It can’t be the average yearly-generated AM, because even the lowest year (2010) is well above this average.

Response: thanks. The data of 3.4 Mt is the average increasing amount per year.

Reviewer #2

1. Having read the revised version of the manuscript, I have the following further comments, compared to my last review:

1) Language quality has been improved but is still not up to the standard of the journal.

Response: Many thanks for your positive assessment. The language has been checked and improved thoroughly.

2) The authors now include estimates of future demand in Fig. 4 and compare them with anthropogenic reserves. Purportedly, this supports their conclusion that anthropogenic materials will be sufficient to cover demand. However, this is not what the comparison shows. Reserves are a cumulative quantity, while demand is a rate. What needs to be compared here are the rate of waste generation and demand. I expect they will not be equal for a long time, particularly under scenarios of increasing demand (which would necessarily accompany increased consumption and increased future waste generation). Furthermore - not all of the material contained in the waste will be recoverable. This is, however, another implicit assumption the authors make without much further discussion. It is unrealistic, particularly for elements like In, Ge, Pt etc. that are highly dispersed in industrial products.

Response: Thanks again for your professional pointing-out. We agreed with you. Based on the equation 10, we revised “anthropogenic reserve” as “anthropogenic mineral” for potential supply of resource generated in each year. Figure 4 has been improved as well. Regarding the recycling rate, some sentences are added to discuss the evolution of recycling imposed by regulation, policy, and technology. We believe the current edition is more scientific and stricter than before. Please check them up.

Fig. 4 Meeting potential of AM for resource consumption of three product groups: (a) Cu; (b) Al; (c) Fe; (d) Plastic;

(e) Rubber; (f) Glass; (g) Pb; (h) Zn; (i) Sn; (j) Mg; (k) Co; (l) Au; (m) Ag; (n) Pd; (o) Rh; (p) Pt; (q) In; (r) W; (s) Nd; (t) Dy; (u) La; (v) other RE; (w) Meeting time of various materials.

We further uncover the supply potential of AM. With the dramatic rise of AM generation and the gradual saturation of material consumption, the potential supply from AM is becoming possible to overtake the resource consumption of three product groups (Fig. 4). Although we are currently still far from a closed-loop society owing to low recycling rate⁷², a rapid advancement is indeed arising for regulation, policy, and technology of circular economy and urban mining. The highly-efficient collection and the cutting-edge recycling will significantly enhance the recycling rate in the future. Thus, if substantial recycling, eighteen materials of AM could meet their demand before 2020, and in 2050 they probably provide over two-fold consumption. The meeting time of Sn and Pd will be approximately 2030 and 2041, respectively. Although Cu, Al, and Co of AM cannot meet their potential consumption by 2050, the disparity gap between their consumption and AMs will be greatly reduced in the following decades. Consequently, AM supply has a growing potential to meet their future resource consumption.

...

Further, the following points could be also concerned: (1) Some specific uses of metals (e.g., indium⁷⁷, aluminum⁷⁸) were more difficult to sufficiently obtain from AM owing to their high dispersion or potential thermodynamic limit⁷⁹. Luckily, continuous endeavors in regulation and technology related to eco-design and the circular economy will hopefully ameliorate this situation;⁸⁰ (2) Investing in research and infrastructure to allow for AM mining can be an effective approach to relieving this resource shortage bottleneck^{22, 81, 82}. Meanwhile, the increase in the amount of AM, as shown by our data, suggests more economic viability in harnessing this reserve; (3) Last, the philosophy of “despising the poor and currying favor with the rich” (prefer to the high-grade mineral), is commonly taken during conventional virgin mining.⁸³ But the environmental and social risks associated with harnessing AM should be fully considered while striving for “treasure” hunting. Both eyes should open for urban mining to reach the double sustainability of the environment and resource.

Other information and data have been updated and improved, please find the full manuscript. MANY THANKS for your valuable comments and suggestions!

Reviewers' comments:

Reviewer #3 (Remarks to the Author):

The authors have adequately addressed the points made by the reviewers at this stage of the revision. Therefore, the manuscript is adequate for publishing.

Reviewer #4 (Remarks to the Author):

General Comment:

The paper hardly a self-standing manuscript. Too much depended upon supplementary information. Some important information is presented in supplementary data and supporting data in the main manuscript and vice versa. Hence, there are the clear scope that reorganization of manuscript in better organized and precise

(i) Not a well-organized and well-written article.

(ii) Inappropriate terminology used, example vehicle and ELV has been used alternatively.

(iii) Lots of scope for vocabulary (inadequate vocabulary) and organizational development, which cannot be managed by a professional English correction service.

(iv) The terminology used is inappropriate provoke question rather answers, like intertwined nexus among... etc.

(v) The introduction need to be rewritten precisely and clearly need to explain what is merit over this communication over existing literature. Advantages need to be explained explicitly and precisely.

(vi) Method: data mining methodology, data dependability need to be addressed.

(vii) Material flow analysis framework is almost empty hardly can sense how MF has been conducted.

(viii) Need to be discussed for better understanding.

(ix) Result and discussion: Discussion about figure 1 and 2 is presented such a way hardly I can understand a thing what really authors want to convey the audience.

Specific comment:

Nomenclatures, abbreviation, Figure, and Table in Supplementary Figure should be separated from each other.

Supplementary Figure 1. It is not presented conceptually correct way. It could be supported by data even.

Supplementary Figure 2: providing not really any important information.

Supplementary Figure 3 and 4: Fig 3 can be explained in 2 lines, no need to dramatize this information. Not at all required. If at all required I would better provide representative figure only.

Supplementary Figure 5: Explain data resources, collection method, accuracy, and dependability, etc. Sometimes in literature data varies quite significantly.

Supplementary Figure 10: Forced fitting reason should be clearly explained.

Supplementary Figure 13A: Is an important inform piece represented at supplementary, I would better consider in the text.

Like this, the review report can be extended to 4-5 pages. Hence, recommendation appended below.

Recommendation: The present manuscript should not be accepted for publication. A complete re-organization, re-writing followed by English vocabulary/grammar/typo correction is essential and recommended.

Reviewer #3

1. The authors have adequately addressed the points made by the reviewers at this stage of the revision. Therefore, the manuscript is adequate for publishing.

Response: Many thanks again for your positive decision. We acknowledge you with the very valuable previous comments.

Reviewer #4

1. General Comment:

The paper hardly a self-standing manuscript. Too much depended upon supplementary information. Some important information is presented in supplementary data and supporting data in the main manuscript and vice versa. Hence, there are the clear scope that reorganization of manuscript in better organized and precise

(i) Not a well-organized and well-written article.

Response: Many thanks for your kind suggestions. All of them are considered in the revised article.

(i) In order to smooth read and well understand, two important figures are moved from supplementary information to the context. One is the designed framework of research methods (original Fig. 7 in supplementary information) as Fig. 1, addressing the detailed and whole thought of this study. The other is the original Supplementary Figure 13A to insert as new Fig. 2, demonstrating all the basic data of anthropogenic minerals.

Fig. 1 The designed framework of research methods in this study. **a** Material flow analysis framework for anthropogenic mineral generation and its boundary in this study; Note: green color for product, and orange color for anthropogenic mineral, and dash line indicates the boundary for this study. **b** Four extract approaches of data collection, data regression, Weibull lifespan curve, and validating methods.

Fig. 2 Estimation of China's AM from 2010 to 2050. **a** WEEE quantity. **b** ELV quantity. **c** WEEE weight. **d** ELV weight. **e** WEEE weight share. **f** ELV weight share. **g** WWC weight. **h** Weight range. **i** Total weight share. **j** Total weight. Note: Large ELV consists of PPV, PVNC, PVC, HCT, MCT, SCT, MiCT, SCar, LCar, LCar, RV, EV, and tractor. Bicycle and MC are classified as small ELV.

2. (ii) Inappropriate terminology used, example vehicle and ELV has been used alternatively.

Response: The terminology is revised from the text to supplementary information. We checked through the article and polish them in a more consistent and accurate expression.

Supplementary Section 1: Terminologies definition and boundary

(1) Solid waste, anthropogenic mineral, and natural mineral¹

Solid waste is any garbage or refuse, sludge from a wastewater treatment plant, water supply treatment plant, or air pollution control facility and other discarded material, resulting from industrial, commercial, mining, and agricultural operations, and from community activities. For instance, municipal solid waste, e-waste, kitchen waste, metal scrap, paper waste, hazardous waste, and animal manure are the typical solid waste.

Anthropogenic mineral (AM), or called urban mineral, belong to solid waste, is the recycled resources characterizing the product with the orderly physical structure and fixed shape under the condition of human activity. In this study, it is defined as the three group of waste products, i.e., waste electrical and electronic equipment, end of life vehicle, and waste wiring & cable.

Natural mineral is the underground solid resource generated by the naturally occurring process. Kaolinite, pyrite, chalcopyrite, and coal are the typical natural mineral.

(2) New scrap and old scrap^{2,3}

New scrap (or preconsumer scrap) is the solid waste originated from the fabrication or manufacturing process. For instance, leftover material, smelting offal, industrial sludge are the typical cases.

Old scrap (or postconsumer scrap) is the solid waste generated from end of life product after consumption or utilization, including the anthropogenic mineral. Waste electrical and electronic equipment, end of life vehicle, waste wiring & cable, spent aircraft, scrap-metal goods, and abandoned buildings belong to old scrap.

(3) Primary resources and secondary resources

Primary resources called natural resources (mineral), are resources that exist without actions of humankind. For instance, oil, iron ore, timber, and coal are the primary resources.

Secondary resources are the resources by anthropogenic activity, which contain some valuable materials like metal within them. For instance, copper scrap, packaging waste, end of life vehicle, and waste oil are the typical cases.

(4) Product, resource, and material⁴

Product is a good consisting of a bundle of tangible and intangible attributes that satisfies consumers.

Resource stands for the useful matters, consisting of valuable material.

Material is the element, constituent, or substance of which something is composed or can be made.

(5) Urban mining and virgin mining⁵

Urban mining is the process recycling resources from anthropogenic mineral, consisting of dismantling, crushing, separation, and deep recovery using hydrometallurgy, pyrometallurgy, or biological treatment.

Virgin mining is the process to extract resources from natural mineral, consisting of crushing, separation, and deep recovery using hydrometallurgy, pyrometallurgy, or biological treatment.

(6) Flow, stock, in-use stock, and net addition to in-use stock (Supplementary Fig. 1)^{4,6}

A flow is the change of a stock over time. The yearly-generated anthropogenic mineral belongs to the flow.

A stock is the reservoir that is accumulated over time by inflows and/or depleted by outflows, which is the totally cumulative quantity.

In-use stock is the cumulative product or resource, which is used or consumed functionally.

Net addition to in-use stock in every year is the difference between the yearly demand and yearly-generated anthropogenic mineral, which is the annual theoretical consumption.

3. (iii) Lots of scope for vocabulary (inadequate vocabulary) and organizational development, which cannot be managed by a professional English correction service.

Response: The vocabulary is added to cover the most words with a professional correction, shown in Supplementary section 2.

Supplementary Section 2: The abbreviations and acronyms of main vocabulary

AC	air conditioner	MCar	medium car
Al	aluminum	MCT	medium cargo truck
Ag	silver	MFA	material flow analysis
AM	anthropogenic mineral	Mg	magnesium
Au	gold	MiCT	mini cargo truck
CDF	cumulative distribution function	MP	mobile phone
Co	cobalt	Mt	million ton
CT	cargo truck	Nd	neodymium
Cu	copper	NPV	New-registration passenger vehicle
DPC	desktop personal computer	Pb	lead
Dy	dysprosium	PC	personal computer
EEE	electrical and electronic equipment	PCV	passenger civil vehicle
ELV	end-of-life vehicle	Pd	palladium
EoL	end of life	PDF	probability density function
Eq.	equation	PPV	private passenger vehicle
Eu	europium	Pt	platinum
EU	European Union	PV	passenger vehicle
EV	electric vehicle	PVC	Private vehicle for civil purpose
e-waste	electrical and electronic waste	RE	rare earth
EWH	electric water heater	RF	refrigerator
Fe	iron	Rh	rhodium
F-C-P	fax machine, copier, and printer	RH	range hood
FM	fax machine	RV	refit vehicle
GWH	gas water heater	SCar	small car
HCT	heavy cargo truck	SCT	small cargo truck
ICT	information and communications technology	SMT	single-machine telephone
In	indium	Sn	tin
La	lanthanum	TV	television
LCar	large car	U.S.	United States
LPC	laptop personal computer	W	tungsten
kg	kilogram	WEEE	waste electrical and electronic equipment
kt	kiloton	WM	washing machine
M	million	WWC	waste wiring and cable
M-CRT	CRT monitor used for mainframe	Y	yttrium
M-FDP	FDP monitor used for mainframe	Zn	zinc
MC	motorcycle		

4. (iv) The terminology used is inappropriate provoke question rather answers, like intertwined nexus among... etc.

(v) The introduction need to be rewritten precisely and clearly need to explain what is merit over this communication over existing literature. Advantages need to be explained explicitly and precisely.

Response: The sentences in the introduction are polished for well understanding.

...

Critical raw materials have also been sinking into AM reserves, while accelerating the depletion of natural minerals.^{12, 13, 14, 15} AM is expected to play an increasing role in resource supply. On the other hand, China is not only the world's major manufacturing power, but also one of the largest consumers and exporter of products.¹⁶ Nowadays China has all types of industries in UN International Standard Industrial Classification System and led in the production of 220 of 500 global industrial products.¹⁷ China in recent years was also the largest importer of secondary resource to alleviate domestic material scarcity. Such features could highlight the unique opportunity for China to uncover the potential of AM supply.^{18, 19}

EEEs and vehicles are the most fashionable aspiration of assets in Chinese households, which are the hallmark of the "Four Big Items" that consumers are aspiring towards (see Supplementary Fig. 2). Their rapid evolution and popularity since the 1970s have led to a dramatic rise in waste accumulation and resource consumption. The consumption of some mineral resources has witnessed multiple increases²⁰, resulting in a shortage of important strategic resources and a growth of external dependence.^{21, 22, 23} To meet future resource consumption, mining from AM has become a global concern and raised the popularization of the concept of a "circular economy".

How to measure the quantity of AM generation and its role in future resource supply is still a crucial scientific challenge. Basic AM information—including generation, composition, and resource flow—is imperative to formulate effective policies making for the recycling industry. At least two gaps can be found in previous studies. First, there is a lack of full discussion for the quantity and quality of China's whole AM reserve and instead only individual types of waste streams have been considered. Zeng et al. (2016) measured the quantity and quality of e-waste, and uncovered China's recycling potential.²⁴ For ELV, van Schaik and Reuter defined the obsolescence rate and recycling rate in the EU,²⁵ and Field et al. (2017) initialed a comprehensive assessment of strategic and minor metals use for passenger cars and light trucks.²⁶ Xue et al. (2013) established the discarding model to examine the recycling potential of ELV.²⁷ Furthermore, the other gap is no publication to accurately measure AM supply meeting potential for the future consumption. The dynamic transfer of existing resources from in-use stock to waste increases both the need for, and possibility of, sustainable resource harvesting from AM.^{28, 29, 30}

Urban metabolism is devoted to facilitate the analysis of the flows of the materials and energy within cities.³¹

Theoretically, the generation and quantity of AM are subject to urban metabolism affected by a variety of regulations, resultant policy, and technological change.^{32, 33} However, due to challenges in tracking and recording, the accurate estimation of AM remains difficult to obtain.³⁴ To complete this study, we collected all the available data and initially created the mathematical models of AM recycling and meeting potential. Four procedures were employed: data collection for the consumption, importation, exportation, and material composition of AM; method development based on material flow analysis (MFA); generation estimation of AM quantity, and resultant economic potential; and validation of results using comparison with previous studies or reported data, sensitivity analysis, and uncertainty analysis (the detailed technical route can be seen in Fig. 1 and Supplementary Fig. 3).

5. (vi) Method: data mining methodology, data dependability need to be addressed.

Response: Figure 1 is added to illustrate research methods. Data dependability is indicated in detail from Supplementary Tables 2-5.

Fig. 1 The designed framework of research methods in this study. **a** Material flow analysis framework for anthropogenic mineral generation and its boundary in this study; Note: green color for product, and orange color for anthropogenic mineral, and dash line indicates the boundary for this study. **b** Four extract approaches of data collection, data regression, Weibull lifespan curve, and validating methods

Supplementary Table 2 I(x) and E(x): Statistics of importation and exportation in China

b Imported AM

Year	WEEE (kt) ¹⁰		Copper scrap (Mt)	Aluminum scrap (Mt)	Steel scrap (Mt)	Plastics scrap (Mt)
	Estimation	Range*	Value	Value	Value	Value
2005			4.82	1.69		
2006			4.94	1.77		
2007			5.58	2.09	3.394	7
2008			5.58	2.15	5.589	7.07
2009			4	2.63	13.69	7.33
2010	877	[600-1500]	4.36	2.85	5.848	8.01
2011	808	[600-1500]	4.69	2.69	6.766	8.38
2012	738	[600-1500]	4.86	2.59	4.974	8.88
2013	669	[600-1500]	4.37	2.50	4.465	7.88
2014	600	600	3.87	2.31	2.564	8.26
2015	531	[0-600]	3.66	2.09	2.33	7.36
2016	462	[0-600]	3.35	1.92	2.16	7.35

Note: data of copper, aluminum, steel, and plastics scrap from National Statistics (<http://data.stats.gov.cn/english/>)

Supplementary Table 3 *w*: Weights of each product and those data distributions used for Monte Carlo simulation

a EEE (kg)¹⁰

Type	Mean	Stand. Dev.	Prob. Dist.	Type	Mean	Stand. Dev.	Prob. Dist.
CRT-TV	27.81	12.01	Beta	MP	0.100	0.023	Beta
CRT-BTV	10.274	1.752	Beta	SMT	0.498	0.114	Beta
RF	40.09	8.79	Beta	RH	23.57	4.21	Beta
MF	9.006	2.681	Beta	FM	6.785	2.437	Beta
LPC	1.858	0.681	Beta	Copier	60.80	37.77	Beta
WM	18.00	3.53	Beta	Printer	6.284	3.448	Beta
CRT-M	13.45	2.58	Beta	EWB	24.36	4.48	Beta
LCD-M	4.885	2.113	Beta	GWH	12.42	2.68	Beta
AC	44.90	20.40	Beta				

Note: LCD-TV: TV with liquid crystal display (LCD); CRT-CTV: color TV with cathode-ray tube (CRT); CRT-BTV: black TV with CRT; RF: refrigerator; MF: mainframe; LPC: laptop personal computer; WM: washing machine; CRT-M: CRT monitor used for mainframe; LCD-M: LCD monitor used for mainframe; AC: air conditioner; DPC: desktop personal computer; MP: mobile phone; SMT: single-machine telephone; RH: range hood; FM: fax machine; EWB: electric water-heater; GWH: gas water-heater.

b Vehicle²¹(ton)

Type	Mean	Range	Prob. Dist.	Type	Mean	Range	Prob. Dist.
LPV	14	[11-16]	Beta	Scar	0.5	[0-1]	Beta
MPV	8	[4-11]	Beta	Mcar	1.2	[1-1.4]	Beta
SPV	3	[2-4]	Beta	Lcar	2.2	[1.4-3]	Beta
MiPV	1	[0-2]	Beta	RV	4	[3-5]	Beta
HCT	16	[14-18]	Beta	EV	1.5	[1-2]	Beta
MCT	10	[6-14]	Beta	MC	0.11	[0.1-0.12]	Beta
SCT	3.4	[1.8-6]	Beta	Tractor	1.6	[1.2-1.9]	Beta
MiCT	0.9	[0-1.8]	Beta	Bicycle	0.015	[0.012-0.018]	Beta

Note: LPV: large passenger vehicle; MPV: medium passenger vehicle; SPV: small passenger vehicle; MiPV: mini passenger vehicle; HCT: heavy cargo truck; MCT: medium cargo truck; SCT: small cargo truck; MiCT: mini cargo truck; Scar: car with capacity ≤ 1L; Mcar: car with capacity of 1-1.6L; Lcar: car with capacity > 1.6L; RV: refit vehicle; MC: motorcycle; EV: electric vehicle.

Supplementary Table 4 c: Average content of resources contained in various products

a WEEE ^{22, 23, 24, 25, 26, 27, 28, 29, 30, 31, 32, 33, 34, 35, 36, 37}

Type	Valuable metals (%)			Precious metals (10 ⁻⁶)			Rare metals (10 ⁻⁵)		Rare earths (10 ⁻⁶)			Plastic (%)	Glass (%) ^d
	Cu	Al	Fe	Au	Ag	Pd	In	Co	Nd	Y	Eu		
RF	3.4	1.1	50	0	0	0	0	0	0	0	0	43.3	2
WM	4	3	53	0	0	0	0	0	0	0	0	26	1
AC	18.5	7	45.9	0	0	0	0	0	0	0	0	17.5	0
CRT-TV	3	2	10	1.4	19.6	0.7	0	0	0	67.51 [*]	5.47 [*]	23	45
LCD-TV	1	4	30	0	0	0	102	0	0	0	0	40	18 ²⁸
CRT-Desktop PC	6.5	2	26	46	207	18.4	0	0	170	4.40E-05 [*]	3.56E-6 [*]	23	40
LCD-Desktop PC	7.2	3.6	18	60	300	25	40	0	270	0	0	4.3	12
LPC	5.7	1.5	20	32	190	19	140	10700	360	0	0	16	20
MP	10.7	2.6	15	25	883	2.6	1102	3738	4500	39	42	25.6	10.6 ³⁹
SMT	2	2	1	2.2	30.8	1.1	10000	0	0	0	0	69	0
FM	4	15	30	0	0	0	0	0	0	0	0	30	0
Copier	3.5	20	16	0	0	0	0	0	0	0	0	35.53	3
Printer	0.5	18	32	0	0	0	0	0	0	0	0	40	0
RH	10	8	30	0	0	0	0	0	0	0	0	5	2
EWH	5	10	35	0	0	0	0	0	0	0	0	5	2
GWH	8	10	30	0	0	0	0	0	0	0	0	5	2

Note: All the data is the average of all collected references; ^aData source from Supplementary Table 5. All the data is assumed to fit for normal distribution.

Tin composition of in e-waste; ^dGlass: 45% for CRT-TV was determined by China's recycler of e-waste recycling, and other data from personal estimation.

b WEEE: Tin content in the typical WEEE

Year	Country	WEEE	RF	WM	AC	TV	MP	Desktop PC	Laptop PC	Hardcopy peripherals	Tin solder weight (ton)	Tin weight in one-ton e-waste (kg)
		Each weight (kg)	40	18	44.9	27	0.1	20.45	1.875	40		
2015	China	Unit (M)	79.93	72.75	153.58	72.98	1969.59	467.14	312.59	62.78	84,952 [*]	3.24
		Weight (Mt)	3.197	1.310	6.896	1.970	0.197	9.553	0.586	2.511		
2016	The U.S.	Unit (M)	10.9	16.41	6.87	39.68	226.17	17.87	45.22	26	3,812	1.05
		Weight (Mt)	0.436	0.295	0.308	1.071	0.023	0.365	0.085	1.040		
Average												2.14

Note: The data of each EEE weigh from Zeng et al. (2016)⁵; China's data for EEs production amount from China Statistics; China's data for EEs ship amount from China Statistics from www.statista.com; ^{*}361.9kt×48.6%×48.3%=84,952 ton⁴⁰; 3,812 from USGS.

c WEEE: Content of other rare earths (Y and Eu) in CRT monitor (w.t. %)

Rare earth	Fluorescent powder							CRT ⁴¹		
	Collected data						Average	CTV	BTV	PC
	Y	12.35 [*]	12.56 [*]	13.53 [*]	13.74 ⁴¹	19.83 ⁴²	19.11 ⁴³	15.19	0.006751111	8.10133E-06
Eu	1.145 [*]	1.13 [*]	1.06 [*]	1.374 ⁴²	1.42 ⁴³	-	1.23	0.000546667	0.000000656	3.56E-10

Note: ^aData determined by Dr Quanyin Tan at Tsinghua University. ^bData determined with the following information: each CRT contains 10-12g fluorescent powder ⁴² in around 27kg, 10kg, and 13.5kg for CRT-CTV, CRT-BTV, and CRT-M, respectively.

d Vehicle: collected data (w.t. %) ^{21, 44, 45, 46, 47, 48, 49, 50, 51, 52, 53, 54}

Resource	Collected data											Mean	Minimum	Maximum
	71	69	64.3	65.4	68	60.6	70.4	68.3	68.76	65.4	67.7			
Fe	71	69	64.3	65.4	68	60.6	70.4	68.3	68.76	65.4	67.7	60.6	67.2	71
Al	4	5	8	7.3	8	5.7	7.0	8.9				8	6.3	4

Cu	1.2	1										1	1.1	1.2
Pb	0.8											0.8	0.8	0.8
Zn	1	1										1	1	1
Plastics	6	7	9.3	9	12.1	10.8	9.1	8.17	9.3	7.8		6	8.86	12.1
Tire & rubber	4	4	5.6	5	5.1	3.42	5.6	4.2				3.42	4.62	5.6
Glass	3	3	2.7	3	2.9	3.1	2.9	2.8				2.7	2.93	3.1

Note: All the data is assumed to fit for normal distribution. The average weight of power battery was 275kg, 235kg, 550kg, and 1900kg for each plug-in passenger vehicle, plug-in commercial vehicle, pure electric passenger vehicle, and pure electric commercial vehicle, respectively (<http://auto.gasgoo.com/News/2017/07/04062820282070016964C501.shtml>).

e Vehicle: composition in various vehicle (%)

Resource	Cu	Al	Fe	Au	Pb	Plastic	Zn	Mg	Rubber	Glass	Co*
PV	1.8	5	69	0.00002	0.8	7	0.05	0.5	5.1	3	-
CT	1.8	5	68	0.00002	0.8	7	0.05	0.5	5.1	3	-
Car	1.8	5	69	0.00002	0.8	7	0.05	0.5	5.1	3	-
RV	1.8	5	68	0.00002	0.8	7	0.05	0.5	5.1	3	-
EV	2	5	68	0.00002	0.8	7	0.05	0.5	5.1	3	0.102
Tractor	1.2	5	68	0	0	7	0.05	0.5	6.6	1	-
MC	2	8	60	0.00002	0	7	0	0	9	3	-
Bicycle	1	10	82	0	0	1	0	0	6	0	-

Note: Data from Supplementary Table 4d and personal estimation. *Since cobalt is used in power battery for EV, its composition in EV will be $0.92\% \times 60\% \times (275+235+550+1900)/(4 \times 4000) = 0.102\%$ [0.92% is the composition of cobalt in MnNiCo cathode material (<http://wemedia.ifeng.com/10681129/wemedia.shtml>); 60% is the cathode material composition in EV battery]²⁵; $(275+235+550+1900)/(4 \times 4000)$ is the average battery weight share in each EV]; Cobalt content in other vehicle is given in Table 4g.

f Vehicle: Mass content of W, Pt, Pd, and Rh in each vehicle (g/car)

Material	Year	Region	Total consumption quantity (ton)	Vehicle share (%)	Vehicle production quantity (k)*	Consumption quantity in each vehicle (g)
W	2000	The U.S.	14,890 ⁵⁵	4.8 ⁵⁵	12,774	56
	2016	World	86,400 ⁵⁶	23 ⁵⁶	94,031	211.3
	Average					134
Pt	2013	World	97.2 ton		86,953	1.118
Pd			77.8 ton			0.895
Rh			21.6 ton			0.248

Note: *data source from https://www.bts.gov/archive/publications/national_transportation_statistics/table_01_23; In 2013, around 97.2 ton, 77.8 ton, and 21.6 ton were globally utilized for Pt, Pd, and Rh, respectively in vehicle industry ([http://www.hysec.com/f/tsnr/\[D2014\]/2014-12/TSNR100/23/RR_3003029292.pdf](http://www.hysec.com/f/tsnr/[D2014]/2014-12/TSNR100/23/RR_3003029292.pdf)), which is used as catalyst to emission controlling. Because the world production quantity was 86,953,000 unit, each vehicle will consume 1.118g Pt, 0.895g Pd, and 0.248g Rh.

g Vehicle: Mass content of other critical metals in an average ELV (g/car)

Metal	min	average	max	Metal	min	average	max	Metal	min	average	max
Dy	1.443256	2.013372	3.23657	Nd	14.06395	18.4593	24.49419	Ag	0.477907	0.725	1.136047
La	24.44767	33.33721	45.83721	Co	23.67442	31.93023	44.88372	Sn (%)	0.0015	0.0019	0.0023

Note: personal calculation based on the data⁵⁷.

h Copper and aluminum consumption in wiring & cable (kt)^{58,59}

Year	1991	1992	1993	1994	1995	1996	1997	1998	1999	2000	2001	2002	2003	2004	2005	2006
Cu	700	780	840	950	1140	1230	1300	1430	1500	1720	2000	2100	2400	2850	3120	3540
Al	220	220	240	250	270	280	320	350	430	580	760	850	930	980	1020	1070

Supplementary Table 5 Resource market prices in recent years (US\$/ton)

Resource	Min	Max	Mean	Error
Ag	0.5×10^6	1.55×10^6	1.03×10^6	$\pm 0.53 \times 10^6$
Al	1.3×10^3	3.3×10^3	2.3×10^3	$\pm 1 \times 10^3$
Au	34×10^6	60×10^6	47×10^6	$\pm 13 \times 10^6$
Co	2.3×10^4	5.2×10^4	3.75×10^4	$\pm 1.45 \times 10^4$
Cu	2.8×10^3	1×10^4	6.4×10^3	$\pm 3.6 \times 10^3$
Dy	2.62×10^5	2.92×10^5	2.77×10^5	$\pm 3 \times 10^4$
Fe	51	188	120	± 68
Glass	26	46	36	± 10
In	13×10^6	35×10^6	24×10^6	$\pm 11 \times 10^6$
La	4.6×10^3	5.4×10^3	6.1×10^3	$\pm 1.4 \times 10^3$
Mg	4×10^3	6×10^3	5×10^3	$\pm 1 \times 10^3$
Nd	6×10^4	1.05×10^5	1.5×10^5	$\pm 9 \times 10^4$
Pb	882	3.6×10^3	2.21×10^3	$\pm 1.36 \times 10^3$
Pd	14×10^6	28×10^6	21×10^6	$\pm 7 \times 10^6$
Plastics*	750	1.03×10^3	890	± 140
Pt	1.9×10^6	5.1×10^6	3.5×10^6	$\pm 1.6 \times 10^6$
Sn	14×10^3	25×10^3	20×10^3	$\pm 5.5 \times 10^3$
Rh	1.3×10^6	3×10^6	2.2×10^6	$\pm 0.9 \times 10^6$
Rubber	160	260	210	± 50
W	6×10^3	1.4×10^4	1×10^4	$\pm 4 \times 10^3$
Y/Eu	190	210	200	± 10
Zn	1.5×10^3	2.8×10^3	2.2×10^3	± 650

Note: Mean is the average of Min and Max. Error = (Max - Min)/2. Data source from London Metal Exchange (<http://www.lme.com>), and InvestmentMine (<http://www.infomine.com>); * Source from six-month operation of a recycling plant in China.

6. (vii) Material flow analysis framework is almost empty hardly can sense how MF has been conducted.

Response: AM is generated with the material flow from consumption to obsolescence. Material flow analysis framework is the core philosophy to examine the AM generation. This philosophy has been adopted to define the mathematical models from equation (1) to equation (9).

7. (viii) Need to be discussed for better understanding.

Response: The logic is polished and more information are added here for better understanding in a fruitful implications. Please check them up.

Three levels are enabled here for the deep discussion. Firstly, the detailed comparison indicated in Supplementary Information Section 4 can verify the results mentioned above, and further consolidate the relevant results (Fig. 2). Thus, China has already overtaken the U.S. to become the world's leading producer of e-waste. All types of cars in weight will exceed the CT and become the leading ELV in 20 years later. In quantity, MC, car, and EV still maintain a rapid increment until 2050. Other vehicles are moving towards saturation (Supplementary Fig. 12). The evolution of WWC weight is subject to longer lifespan of wiring and cable than those of EEE and vehicle. Rapid urbanization and economic growth in 1980s-2010 will impose a significant influence after 2025. China has revised the standard of lifespan for wiring and cable from 20 years to 70 years, which can decline the obsolescence of wiring and cable. Lastly, the circular urged the related ministries and departments to continue cracking down on smuggling of dangerous trash, medical waste, e-waste, and household garbage (Supplementary Table 10).^{62, 73} The importation of AM will fall dramatically so that there will perhaps be a couple of years for the whole disappear of importation (Fig. 2e).

Secondly, natural minerals are encountering a couple of the tremendous challenges. Au, Ag, and In in global remaining underground reserve can only sustain the future consumption for 36 years, 9 years, and 4 years at half of U.S. per capita consumption rates.⁷⁴ Their exploration and mining will be far harder and more costly owing to the increasing depth of exploration and the descending grade.^{75, 76} Most anthropogenic base metal minerals have the higher grade and purity (as the pure metal or alloy) than the general natural minerals. Eventually, through a circular economy AMs will play an essential role for supply security and for overall sustainability objectives. China's focus will be shifting from the dependence of virgin mining and waste imports to domestic urban mining of AM.

Lastly, to improve urban mining and circular economy, the following points could be also concerned: (1) Some specific uses of metals (e.g., indium⁷⁷, aluminum⁷⁸) were more difficult to fully obtain from AM owing to their high dispersion or potential thermodynamic limit⁷⁹. Luckily, continuous endeavors in regulation and technology related to the circular economy and zero waste will hopefully ameliorate this situation;⁸⁰ (2) Investing in research and infrastructure to allow for AM mining can be an effective approach to relieving this resource shortage bottleneck^{22, 81, 82}. Meanwhile, the increase in the amount of AM, as shown by our data, suggests more economic viability in harnessing this reserve; (3) Last, the philosophy of "despising the poor and currying favor with the rich" (prefer to the high-grade mineral), is commonly taken during conventional virgin mining.⁸³ But the environmental and social risks associated with harnessing AM should be fully considered while striving for "treasure" hunting. Both eyes

should open for urban mining to reach the double sustainability of the environment and resource.

8. (ix) Result and discussion: Discussion about figure 1 and 2 is presented such a way hardly I can understand a thing what really authors want to convey the audience.

Response: Figure 1 is improved to illustrate the AM generation and evolution. Figure 2 is to uncover the evolution and map of contained resources in AM.

9. Specific comment:

Nomenclatures, abbreviation, Figure, and Table in Supplementary Figure should be separated from each other.

Supplementary Figure 1. It is not presented conceptually correct way. It could be supported by data even.

Supplementary Figure 2: providing not really any important information.

Supplementary Figure 3 and 4: Fig 3 can be explained in 2 lines, no need to dramatize this information. Not at all required. If at all required I would better provide representative figure only.

Supplementary Figure 5: Explain data resources, collection method, accuracy, and dependability, etc. Sometimes in literature data varies quite significantly.

Supplementary Figure 10: Forced fitting reason should be clearly explained.

Supplementary Figure 13A: Is an important inform piece represented at supplementary, I would better consider in the text.

Response: Thanks again for your professional pointing-out.

(1) Okay, we separate them well.

(2) Nomenclatures are enriched again for better understanding without Supplementary Figure 1.

(3) Supplementary Figure 2 helps to understand the difference of the production, consumption, and theoretical consumption. It is also suggested from another reviewer to add.

(4) Okay, we keep Supplementary Figure 3 and shorten the notes.

(5) Supplementary Figure 5 (Figure 3 in new Supplementary document) is devoted to illustrate the whole technical route of this study with new Figure 1 in new revised article.

(6) Supplementary Figure 10 is explained from the text to the notes with the theory and references.

(7) Agree, the information in Supplementary Figure 13A has been combined as new Figure 2 in the new revised article.

Reviewers' comments:

Reviewer #4 (Remarks to the Author):

Title: Anthropogenic mineral supply through a circular economy approach has potential to meet Chinese resource consumption

General Comment:

Still, the manuscript is far from the acceptance for publication. The justifications are given below.

Title is a statement: Yes, every circular economist, recycling researcher, and policymaker knew it well. Hence, the title needs to tell the story of the subject. The subject is the author has developed a modeling tool to predict the future of AM generation and its potential to meet demand from secondary resources. Hence, the title is not reflecting the story rather a statement concludes a very subjective concept. Revising title recommended.

Abstract has spent over 2 pages to start the subject itself, which is an ineffective way of presenting. The abstract is lacking to tell what are the challenges are rather a long story of AM which is not precise. I would tell combine first 4 paragraph into a paragraph and rather jump to problem and issues, what is the approach of paper to address the issue.

I do not find really why this particular model is important to consider and associated benefits. In my understanding data has the potential to justify it. There is lots of scope in the data to develop real-world confidence which has been neglected.

Real-world benefit of the model is not quite clear. Of course, one can very well be correlated based on data, which authors are lacking to explain well.

In the data treatment significance digit is not fully taken into consideration. Example Figure 2(a), hence, should be taken care of.

To justify the model actually works within a certain time limit, some proof should be given. Justifying the model could be true with 95% confidence interval etc.

I believe there is lots of scope in the data to correlate how the model can address the AM circular economy issue in particular and related issues in general.

Recommendation: The manuscript is not ready to be accepted.

Reviewer #4

1. Title is a statement: Yes, every circular economist, recycling researcher, and policymaker knew it well. Hence, the title needs to tell the story of the subject. The subject is the author has developed a modeling tool to predict the future of AM generation and its potential to meet demand from secondary resources. Hence, the title is not reflecting the story rather a statement concludes a very subjective concept. Revising title recommended.

Response: Many thanks for your suggestions. Actually, in the past two years the title has revised several times. After new discussion, the title is improved as:

Mapping anthropogenic mineral generation in China and its implications for a circular economy

2. Abstract has spent over 2 pages to start the subject itself, which is an ineffective way of presenting. The abstract is lacking to tell what are the challenges are rather a long story of AM which is not precise. I would tell combine first 4 paragraph into a paragraph and rather jump to problem and issues, what is the approach of paper to address the issue.

Response: Very good suggestions. Thanks. The abstract is greatly shortened to highlight the problem and contribution.

Abstract: Anthropogenic mineral is absorbing wide concern in the context of circular economy, but its generation mechanism and quantity from product to waste remain unclear. Here we consider three product groups, 30 products, and use the revised Weibull lifespan model to map the recycling potential of anthropogenic mineral and 23 types of the capsulated materials by targeting their evolution from 2010 to 2050. Total weight of anthropogenic mineral on average in China reached 39 Mt in 2010, but it will double in 2022 and quadruple in 2045. Stocks of precious metals and rare earths will increase faster than most base materials. The total economic potential in yearly-generated anthropogenic mineral is anticipated to grow markedly from 100 billion US\$ in 2020 to 400 billion US\$ in 2050. Anthropogenic mineral of around 20 materials will be able to meet projected consumption of three product groups by 2050, due to high availability of recycled content and gradual saturation of consumption.

3. I do not find really why this particular model is important to consider and associated benefits. In my understanding data has the potential to justify it. There is lots of scope in the data to develop real-world confidence which has been neglected. Real-world benefit of the model is not quite clear. Of course, one can very well be correlated based on data, which authors are lacking to explain well.

Response: Some sentences are improved to highlight the feasibility and applicability of the model. The revised Weibull distribution model is designed with a full consideration of regulated lifespan. Please check. The main innovation and contribution cover two big points: one is that all the anthropogenic mineral generation from 2010 to 2050 in China (Figs. 2-4) is firstly measured with material flow analysis framework and Weibull lifespan distribution model, whose parameters are fully defined, in particular for vehicle (Supplementary Table 12), with a time-consuming review of global literature, report, and some interviews; The second is to measure the relationship between anthropogenic mineral reserve and future material demand (Fig. 5). Real-world situation in China has been considered from the quantity of product to Weibull lifespan distribution parameters. Additional explanation is added for better understanding.

Material flow analysis framework. *More than ten methods or models have been adopted for AM estimation in previous studies (Their applications and difference can be seen in Supplementary Tables 6 and 7). The selection of a particular method depends mainly on data availability and robustness.^{38, 39} During the process of urban metabolism, products flow into the society (sales), then accumulate in the built environment (stock); when reaching end of life (EoL) after a certain period (lifespan), they flow out as an AM.⁴⁰ MFA models quantitatively describe the dynamics, magnitude and interconnection of product sales, stocks and lifespans.^{41, 42, 43} ...*

Data regression. *Any given products as the net production from Eq. 1 was assumed to be both consumed and discarded in China, and its future demand was then determined using a time-step method based on data regression (Fig. 1). Ideally, the Logistic equation is feasible and applicable to fulfill the forecasting future consumption. It is characterized of the rapid growth at the start stage, stable growth at the middle stage, and slow growth for a constant at the late stage, and could be expressed by exponential function, linear function, and constant value, respectively.⁴⁴ Additionally, the decline of production and consumption commonly occurs while the product is replaced. This principle is substantially applied to align the unavailable data.*

...In particular, the Weibull statistical distribution was chosen for this study to model the lifetime of product (Fig. 1). For non-regulated-lifespan products like EEEs and bicycle, the probability density function (PDF) of the Weibull distribution is given by Eq. 2.^{51, 56} Regarding the regulated-lifespan products like vehicle, and wiring and cable, the regulated lifespan should be considered for a revised Weibull lifespan curve (Eq. 3).

Supplementary Fig. 13 The comparison of typical AM estimation

a Typical WEEE quantity; **b** Typical WEEE weight; **c** Typical ELV quantity. Note: Typical WEEE covered RF, WM, TV, PC, and AC; typical ELV covered passenger vehicle, cargo truck, car, and refit vehicle. Yang et al. (2008)¹³, Li et al. (2006)¹⁴, Liu et al. (2006)¹⁵, Veenstra et al. (2010)¹⁶, Baldé et al. (2015 & 2017)^{17,18}, Duan et al. (2016)¹⁹, Xi et al. (2014)²⁰, Xue et al. (2013)²¹, and Jin et al. (2008)²².

Supplementary Fig. 14 Validation for all the registered vehicles quantity

a Comparison of estimated value in this study to real value; **b** significance test for real value and estimated value.

4. In the data treatment significance digit is not fully taken into consideration. Example Figure 2(a), hence, should be taken care of.

Response: thanks, the significance digit of all the relevant data has been checked and improved in the two-separating document as **Source Data** and **Supplementary excel table**. Here are some instances.

5. To justify the model actually works within a certain time limit, some proof should be given. Justifying the model could be true with 95% confidence interval etc. I believe there is lots of scope in the data to correlate how the model can address the AM circular economy issue in particular and related issues in general.

Response: Some proof has been added to validate the results. The discussion to reflect the real-world situation is also improved especially for the obtained results. Supplementary documents have been polished again.

...Data regression is enabled with Supplementary Figs. 7-9 for the annual net production and imported scraps until 2050. In this year China was attempted to become one moderately developed country...

.....

Deeper discussion is enabled here at three levels of obtained results, potential implications, and recommendation. Firstly, the detailed comparison from the previous studies to real-world data was indicated in Supplementary Information Section 4, which can verify the obtained results, and further consolidate the relevant results (Fig. 2). Thus, China has already overtaken the U.S. to become the world's leading producer of e-waste since 2014.²⁴ MC, car, and EV in quantity will maintain a rapid increment from 2015 to 2050. China provided half of global EV stock with a selling of 1.1 million in 2018.⁶⁹ Other vehicles are moving towards saturation (Supplementary Fig. 10). The evolution of WWC weight is subject to longer lifespan of wiring and cable than those of EEE and vehicle. Rapid urbanization and economic growth in 1980s-2010 will impose a significant influence on AM generation after 2025. China has revised the standard of lifespan for wiring and cable from 20 years to 70 years, which can decrease the obsolescence of wiring and cable. Additionally, the Chinese government has issued strict mandates to continue cracking down on smuggling of dangerous trash, medical waste, e-waste, and household garbage (Supplementary Table 10).^{58, 70} The importation of AM will fall dramatically leading to eventual cessation of such external sources within a few years (Fig. 2e). Around 99% plastic waste, for instance, has been descended from 7.05 Mt in 2017 to 0.076 Mt in 2018. The solution to plastic waste management is becoming an increasing concern in the European countries, like the UK and France.

REVIEWERS' COMMENTS:

Reviewer #3 (Remarks to the Author):

By following the answers provided by the authors about the comments of Reviewer #4, I have followed very carefully every answer.

The authors have answered the five comments with a detailed explanation and have added important information so that each doubt will be completely satisfied. In addition, the authors have put emphasis on the source data which has been included as an Excel supplementary document.

With these arguments, and taking into account that the authors have put a lot of effort to improve the manuscript, I consider it deserves to be published in Nature Communications.

Reviewer #3 (Remarks to the Author):

By following the answers provided by the authors about the comments of Reviewer #4, I have followed very carefully every answer.

The authors have answered the five comments with a detailed explanation and have added important information so that each doubt will be completely satisfied. In addition, the authors have put emphasis on the source data which has been included as an Excel supplementary document.

With these arguments, and taking into account that the authors have put a lot of effort to improve the manuscript, I consider it deserves to be published in Nature Communications.

Response: Many thanks for your valuable comments and efforts.